# FAK activity in cancer-associated fibroblasts is a prognostic marker and a druggable key metastatic player in pancreatic cancer

Sonia Zaghdoudi[1,†], Emilie Decaup[1,†], Ismahane Belhabib[1], Rémi Samain[1], Stéphanie Cassant-Sourdy[1], Julia Rochotte[1], Alexia Brunel[1], David Schlaepfer[2], Jérome Cros[3], Cindy Neuzillet[4], Manon Strehaiano[1], Amandine Alard[1], Richard Tomasini[5], Vinothini Rajeeve[6], Aurélie Perraud[7], Muriel Mathonnet[7], Oliver MT Pearce[8], Yvan Martineau[1] iD, Stéphane Pyronnet[1], Corinne Bousquet[1,*] iD & Christine Jean[1,**] iD

## Abstract

Cancer-associated fibroblasts (CAFs) are considered the most abundant type of stromal cells in pancreatic ductal adenocarcinoma (PDAC), playing a critical role in tumour progression and chemoresistance; however, a druggable target on CAFs has not yet been identified. Here we report that focal adhesion kinase (FAK) activity (evaluated based on 397 tyrosine phosphorylation level) in CAFs is highly increased compared to its activity in fibroblasts from healthy pancreas. Fibroblastic FAK activity is an independent prognostic marker for disease-free and overall survival of PDAC patients (cohort of 120 PDAC samples). Genetic inactivation of FAK within fibroblasts (FAK kinase-dead, KD) reduces fibrosis and immunosuppressive cell number within primary tumours and dramatically decreases tumour spread. FAK pharmacologic or genetic inactivation reduces fibroblast migration/invasion, decreases extracellular matrix (ECM) expression and deposition by CAFs, modifies ECM track generation and negatively impacts M2 macrophage polarization and migration. Thus, FAK activity within CAFs appears as an independent PDAC prognostic marker and a druggable driver of tumour cell invasion.

**Keywords** cancer-associated fibroblasts; extracellular matrix remodelling; focal adhesion kinase; metastasis; pancreatic ductal adenocarcinoma
**Subject Categories** Biomarkers; Cancer

## Introduction

Pancreatic ductal adenocarcinoma (PDAC) has a poor prognosis among carcinomas, with 5-year survival rates < 7% due to a late diagnosis and frequent metastases (50% of cases at diagnosis). Tumour resection is possible only in non-metastatic patients with locally non-advanced tumour, but operated patients still have a life expectancy of < 5 years (Siegel *et al*, 2017). Chemotherapies have provided only limited survival benefit, although they show cytostatic effects in pre-clinical PDAC models (Hidalgo, 2012). Therapeutic resistance may in part be explained by the high amount of desmoplasia within the tumour microenvironment which represents up to 80% of the tumour mass (Feig *et al*, 2012). This fibrotic environment is characterized by increased deposition and altered organization with enhanced stiffening of extracellular matrix (ECM). ECM can form a physical barrier to drug delivery (Olive *et al*, 2009), contribute to the creation of a hypoxic environment that selects for more resilient cancer cells (Hwang *et al*, 2008), and also influences immune cell trafficking into and within the tumour (Hallmann *et al*, 2015). Both ECM composition and architecture are altered in carcinomas (Provenzano *et al*, 2006), and such changes promote tumour

1 Cancer Research Center of Toulouse (CRCT), team 6 "Protein synthesis & secretion in carcinogenesis", Equipe labellisée Ligue Contre Le Cancer, Labex TOUCAN, INSERM UMR 1037- University Toulouse III Paul Sabatier, Toulouse, France
2 Department of Reproductive Medicine Moores Cancer Center, University of California San Diego, La Jolla, CA, USA
3 Department of Pathology, Beaujon Hospital, INSERM U1149, Clichy, France
4 Medical Oncology Department, Curie Institute, Versailles Saint-Quentin University, Saint-Cloud, France
5 Cancer Research Center of Marseille, INSERM U1068, Marseille, France
6 Centre for Genomics and Computational Biology, Barts Cancer Institute, Queen Mary University of London, London, UK
7 EA 3842 Laboratory, Medicine and Pharmacy Faculties, Limoges University, Limoges, France
8 Centre for Tumour Microenvironment, Barts Cancer Institute, London, UK
*Corresponding author. Tel: +33 582741653; E-mail: corinne.bousquet@inserm.fr
**Corresponding author. Tel: +33 582741650; E-mail: christine.jean@inserm.fr
†These authors contributed equally to this work

progression via the generation of ECM tracks used by tumour cells to invade (Gaggioli *et al*, 2007). Altogether, ECM provides critical chemical and mechanical cues promoting the acquisition and maintenance of each of the cancer hallmarks and its abundance, organization and stiffness modulate every behavioural facet of tumour cells (for review, see (Pickup *et al*, 2014)). ECM synthesis, maturation and organization are largely regulated by fibroblasts (Erdogan & Webb, 2017).

Cancer-associated fibroblasts (CAFs), the most abundant PDAC stromal cells, are important promoters of tumour growth, disease progression and resistance to therapies (Pietras & Ostman, 2010; Ostman, 2012). Fibroblast activation correlates with PDAC poor prognosis and tumour recurrence after surgery (Erkan *et al*, 2012). CAFs increase tumour aggressiveness (survival, invasion, chemoresistance) via the secretion of growth factors, cytokines and chemokines (Egeblad *et al*, 2010). CAFs also impact other stromal cells promoting angiogenesis with immature permeable vessels, inflammation and immune suppression (Sun *et al*, 2018a). Moreover, in several epithelial cancers, CAFs were reported to produce and deposit elevated amounts and abnormal varieties of ECM components (Tuxhorn *et al*, 2002; Schor *et al*, 2003; Erdogan & Webb, 2017). Overall, CAFs and abnormal ECM deposition play critical roles in PDAC progression, deleterious for patient survival.

A key intracellular effector of ECM signalling is focal adhesion kinase (FAK), which dimerizes and gets activated upon ECM-induced integrin receptor activation. This leads to FAK autophosphorylation at residue Y397, binding of SRC-family kinases and subsequent phosphorylation of the FAK domain activation loop (Y576 and Y577) (Sulzmaier *et al*, 2014). FAK is a cytoplasmic protein tyrosine kinase controlling cell movement, invasion, survival and gene expression through its kinase or scaffolding functions. In advanced-stage solid cancers, FAK is often over-expressed and activated in tumour cells thus fostering tumour progression and metastasis (Sulzmaier *et al*, 2014). Therapeutic benefits of pharmacological FAK inhibition in preclinical models (Hochwald *et al*, 2009) have been thus far allocated to the direct role of FAK expressed by neoplastic cells. However, one can assume that pharmacological FAK inhibitors will also target tyrosine kinase activity in CAFs. It was previously described that specific FAK inactivation in endothelial cells prevented spontaneous tumour metastasis by enhancing barrier function (Jean *et al*, 2014). Whereas the role of FAK activity is largely documented in normal fibroblasts (for review see (Schlaepfer & Mitra, 2004)), very little is known about the role of FAK in activated fibroblasts present in a tumour context (Barker *et al*, 2013).

We therefore evaluated FAK activity in CAFs from human PDAC and deciphered its implication in PDAC progression, invasion and metastasis. We found that FAK activity within CAFs (using the phosphorylation level of FAK on tyrosine (Y) 397 as readout) is an independent prognostic marker for disease-free survival (DFS) and overall survival (OS) in PDAC patients. Functionally, we also revealed that FAK activity within PDAC CAFs plays a prominent role in tumour cell metastasis and in intra-tumour immune cell recruitment in correlation with its regulatory impact on ECM synthesis and deposition by CAFs. PDAC patients may therefore benefit from FAK targeting though inhibition of the CAF functions.

# Results

## Increased FAK activity in human pancreatic adenocarcinoma-associated fibroblasts is an independent prognostic marker

We first sought to determine whether fibroblastic FAK activity was associated with PDAC patient outcome. The level of FAK autophosphorylation at tyrosine 397 (pY397 FAK) was quantified, as a readout of FAK activity, by immunohistochemistry (IHC) on a human tissue array composed of 40 PDAC tumours and 10 healthy human pancreas (Fig 1A and B, high and low magnification, respectively; see Appendix Fig S1A–C for antibody validation). Expression of α-SMA and FAP-α was used as surrogate markers of CAFs (fibroblasts from PDAC but not from healthy pancreas were positive; Fig 1A and B). We observed a very low or null FAK activity in fibroblasts from normal control tissues (identified based on their particular spindle-shaped morphology), whereas fibroblasts from PDAC show an important and significant increase of pY397 FAK staining compared to controls in the whole cell, as well as in the cytoplasm and in the nucleus (Fig 1A–C). Using the average of fibroblastic FAK activity quantified in healthy tissues as a baseline, fibroblastic FAK activity appears increased in 87.5% (35/40) of PDAC patients.

Given the patient heterogeneity found for fibroblastic FAK activity in this discovery cohort (Fig 1C), we have challenged the prognostic value of FAK activity within CAFs on a human tissue array of an independent retrospective cohort composed of 120 resected PDAC samples. Clinical and pathological data from those patients are summarized in Table EV1. At the time of last follow-up, tumour relapse was reported in 80 patients (66.6%), and 67 patients (55.8%) had died. Median DFS and OS were 13.1 months (range: 1.0–113 months) and 19.69 months (range: 2.6–113 months), respectively. Positive resection margin (R1) and lymph node ratio (LNR) > 0.20 were both significantly associated with shorter DFS and OS in univariate analysis (Table 1).

In order to calculate a pY397 FAK score as a surrogate of FAK activity in each sample, and quantify its expression in stromal vs tumoural area, we used the automatic computer-assisted

---

**Figure 1.** **Increased FAK activation in human pancreatic adenocarcinoma-associated fibroblasts.**

A, B  Representative haematoxylin and eosin (H&E) staining, and immunohistochemistry of phosphorylated FAK on tyrosine (Y) 397 (pY397 FAK), alpha-smooth muscle actin (α-SMA) and fibroblast activated protein alpha (FAP-α) in serial sections of human normal pancreas (Ct1), of human normal adjacent PDAC tissue (Ct2) and of human PDAC tissues (PDAC1-3); scale bar, 100 μm (A), and 300 μm (B).

C  Quantification of pY397 FAK level in cancer-associated fibroblasts from human PDAC microarray: 40 different PDAC patient samples and 10 normal human pancreas have been stained for pY397 FAK and double blinded quantified (based on intensity and positive cell number) on a scale between 0 (no staining) and 6 (Y397 FAK highly positive) in whole cell (left), cytoplasm (middle) and nucleus (right). Values are means ± SEM. \*\*$P < 0.01$, by unpaired two-tailed Student's *t*-test.

Source data are available online for this figure.

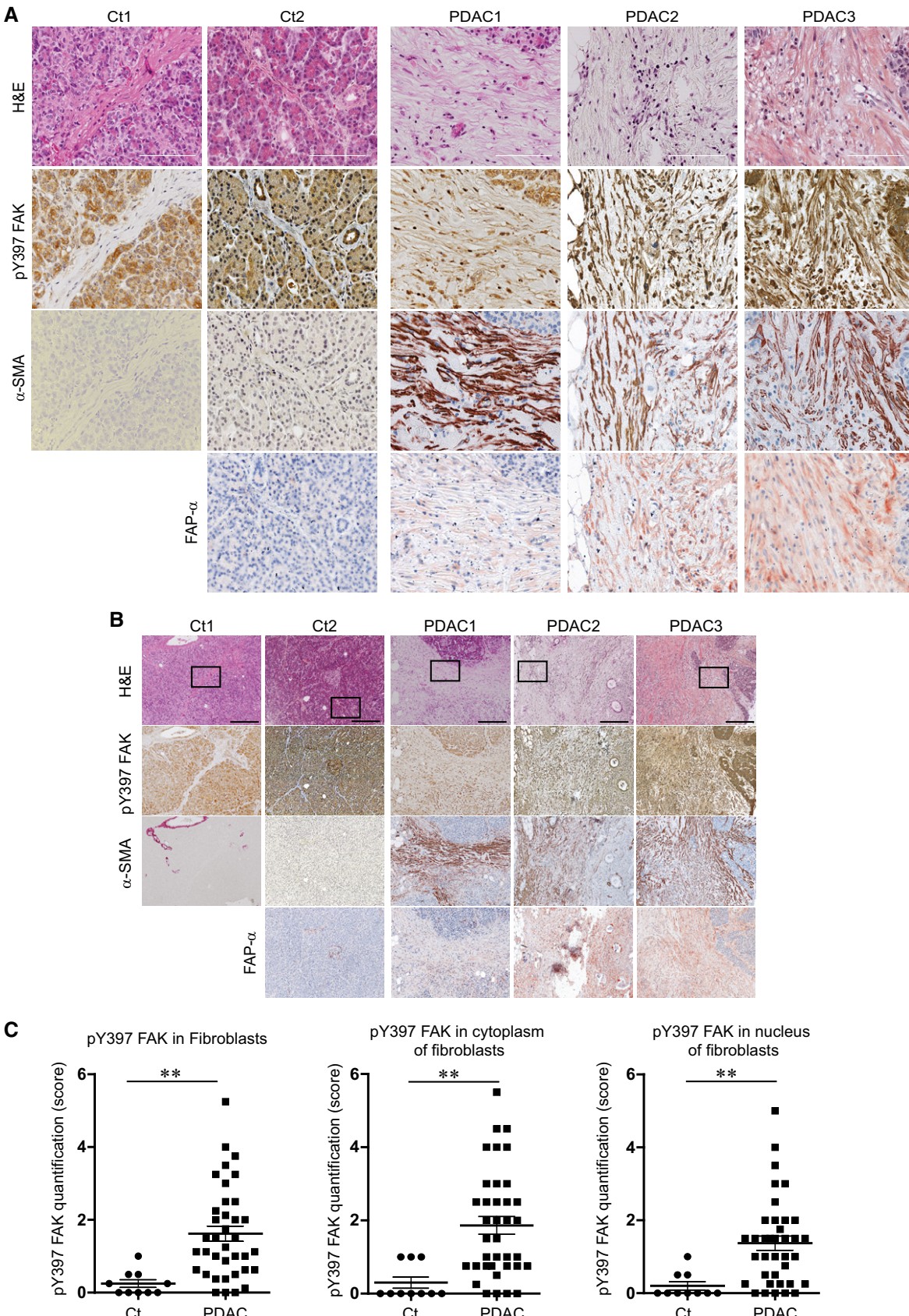

**Figure 1.**

Table 1. Correlation between pathological data and pY397 FAK score in CAFs with survival: results of univariate and multivariate analyses in the 120 patient cohort (only variables with *P* < 0.05 are presented).

|  | DFS | | OS | |
|---|---|---|---|---|
|  | HR (95% CI) | *P* | HR (95% CI) | *P* |
| Univariate | | | | |
| Positive resection margin. R1 | 2.096 (1.192–3.685) | 0.01 | 2.004 (1.070–3.753) | 0.030 |
| LNR (ration N+/N) > 0.20 | 2.200 (1.296–3.734) | 0.003 | 2,122 (1.178–3.823) | 0.012 |
| High fib. pY397 FAK score | 1.834 (1.167–2.880) | 0.008 | 1.929 (1.156–3.218) | 0.012 |
| Multivariate | | | | |
| Positive resection margin. R1 | 1.711 (1.037–2.822) | 0.036 | 1.874 (1.058–3.321) | 0.031 |
| LNR (ration N+/N) > 0.20 | 1.779 (1.097–2.886) | 0.020 | 2.102 (1.189–3.718) | 0.011 |
| High fib. pY397 FAK score | 1.616 (1.017–2.566) | 0.042 | 1,898 (1.133–3.180) | 0.015 |

CI, confidence of interval; DFS, disease-free survival; HR, hazard ratio; LNR: lymph node ratio; OS, overall survival.

analyser Definiens Tissue Studio®. Appendix Fig S2A and B validate that Definiens® software properly discriminates CAFs (α-SMA- and/or FAP-α-positive cells) from other cells (including isolated tumour cells, positive for pan-cytokeratin) based on cell morphology. We then applied Definiens® software on our tissue array previously stained for pY397 FAK (Fig 2A). Quantification of pY397 FAK scores shows that, in the whole cohort (*n* = 120), high FAK activity within CAFs (60/120) does not correlate with any of the tested variables (known prognostic factors), including resection margin status (R0 vs R1), tumour size, TNM stage and LNR (ratio of positive lymph nodes over total dissected lymph nodes; Table 1). However, using univariate analysis, the FAK activity score correlates with patient survival, where patients with high FAK activity within CAFs have significantly shorter DFS (13.5 vs 19.1 months, HR: 1.834, *P* = 0.0085) and shorter OS (26.3 vs 36.1 months, HR: 1.921, *P* = 0.0119; Table 1; Fig 2B and C). When stratified based on the grade status, fibroblastic FAK activity is predictive of shorter DFS and OS for patients with low and moderate grade but not for patients with high grade tumours (Fig EV1A–D). Across all patient samples, positive resection margin, LNR > 0.20, and high fibroblastic pY397 FAK score were independently associated with DFS and OS in multivariate analysis (Table 1). Conversely, FAK activity in the tumour cell compartment had no prognostic impact on DFS (14.10 vs 15.67 months, HR: 1.112, *P* = 0.958) nor OS (32.98 vs 33.51 months, HR: 1.016, *P* = 0.7549), (Fig 2D–E).

Together these results indicate that FAK is highly activated in human pancreatic adenocarcinoma-associated fibroblasts. High fibroblastic FAK activity (pY397 FAK score) in PDAC tumours is an independent prognostic factor for DFS and OS in multivariate analysis.

## Fibroblastic FAK inactivation prevents tumour metastasis without changing tumour growth

Since we had found that fibroblastic FAK activity in human PDAC correlates with accelerated relapse and shorter survival, we explored its functional role on PDAC progression using a knock-in point mutation strategy (FAK lysine 454 to arginine) in *fak* exon 21 to generate fibroblasts expressing either a kinase inactive (kinase-dead) FAK (FAK-KD) or active (wild-type) FAK (FAK-WT) (Lim *et al*, 2010). Mouse fibroblasts expressing FAK-WT or FAK-KD were orthotopically and syngeneically co-grafted with mouse pancreatic tumour cells (isolated from LSL-K-ras$^{G12D/+}$; p53$^{R172H/+}$; Pdx1-Cre, also referred as KPC) in C57BL/6 immunocompetent mice. From our previous observations (Duluc *et al*, 2015), co-grafting of tumour cells with fibroblasts slightly increases tumour growth compared to tumour cells grafted alone, which was also seen with FAK-KD fibroblasts (Fig 3A). To further validate that FAK-WT and FAK-KD fibroblasts are activated into CAF-like cells once co-grafted with tumour cells, GFP-expressing fibroblasts were co-injected with non-labelled tumour cells, and CAF markers α-SMA, FAP-α and PDGFR-α evaluated. At the time of sacrifice, the level of FAK pY397 is highly decreased in FAK-KD fibroblasts compared to FAK-WT, as expected (Fig EV2A). Importantly, GFP-positive cells, from both groups, express high levels of α-SMA, FAP-α and PDGFR-α (Fig EV2B), confirming the activated status of the grafted fibroblasts. Interestingly, most cells expressing α-SMA, FAP-

---

Figure 2. Increased FAK activation in human pancreatic adenocarcinoma-associated fibroblasts is an independent prognostic marker.

A   Representative images of Definiens® Tissue Studio multi-resolution segmentation algorithm on two PDAC TMA examples (X, Y). From left to right: IHC for pY397 FAK; ROI detection of tumour cells (dark blue), stroma (orange) and other (light blue/grey); nuclear/cytoplasm detection in blue and green, respectively; IHC staining classification for pY397 FAK intensity signal (dark red for high signal, orange for medium signal, yellow for low signal and white for negative signal).

B, C   Disease-free survival (B) and overall survival (C) from 120 patient cohort according to pY397 FAK expression in CAFs.

D, E   Disease-free survival (D) and overall survival (E) from 120 patient cohort according to pY397 FAK expression in tumour cells.

Data information: (B–E) Survival curves were estimated with the Kaplan–Meier method and compared using log-rank test, non-parametric tests (chi-square test) to compare independent groups for categorical data; HR: hazard ratio.

Source data are available online for this figure.

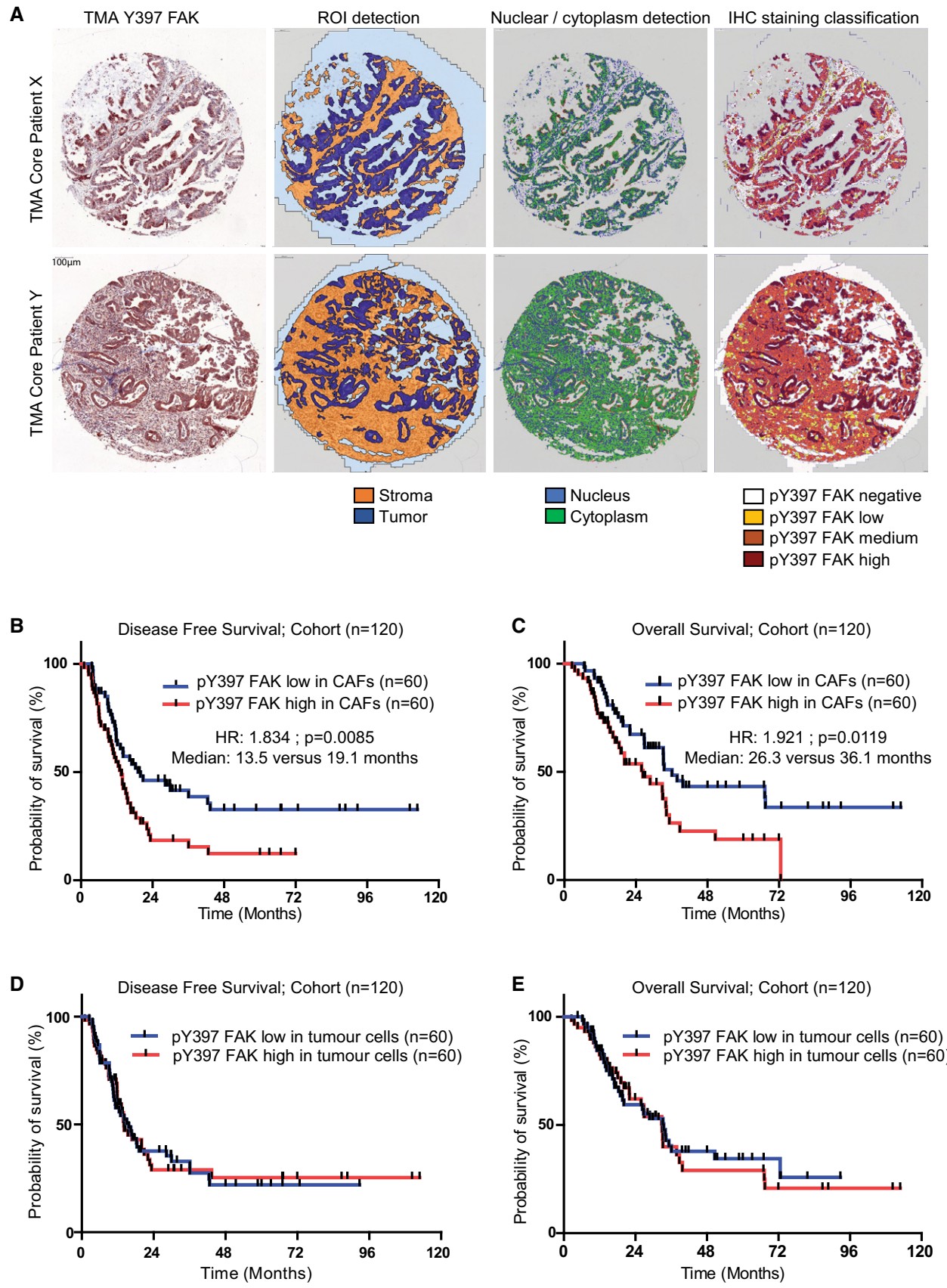

Figure 2.

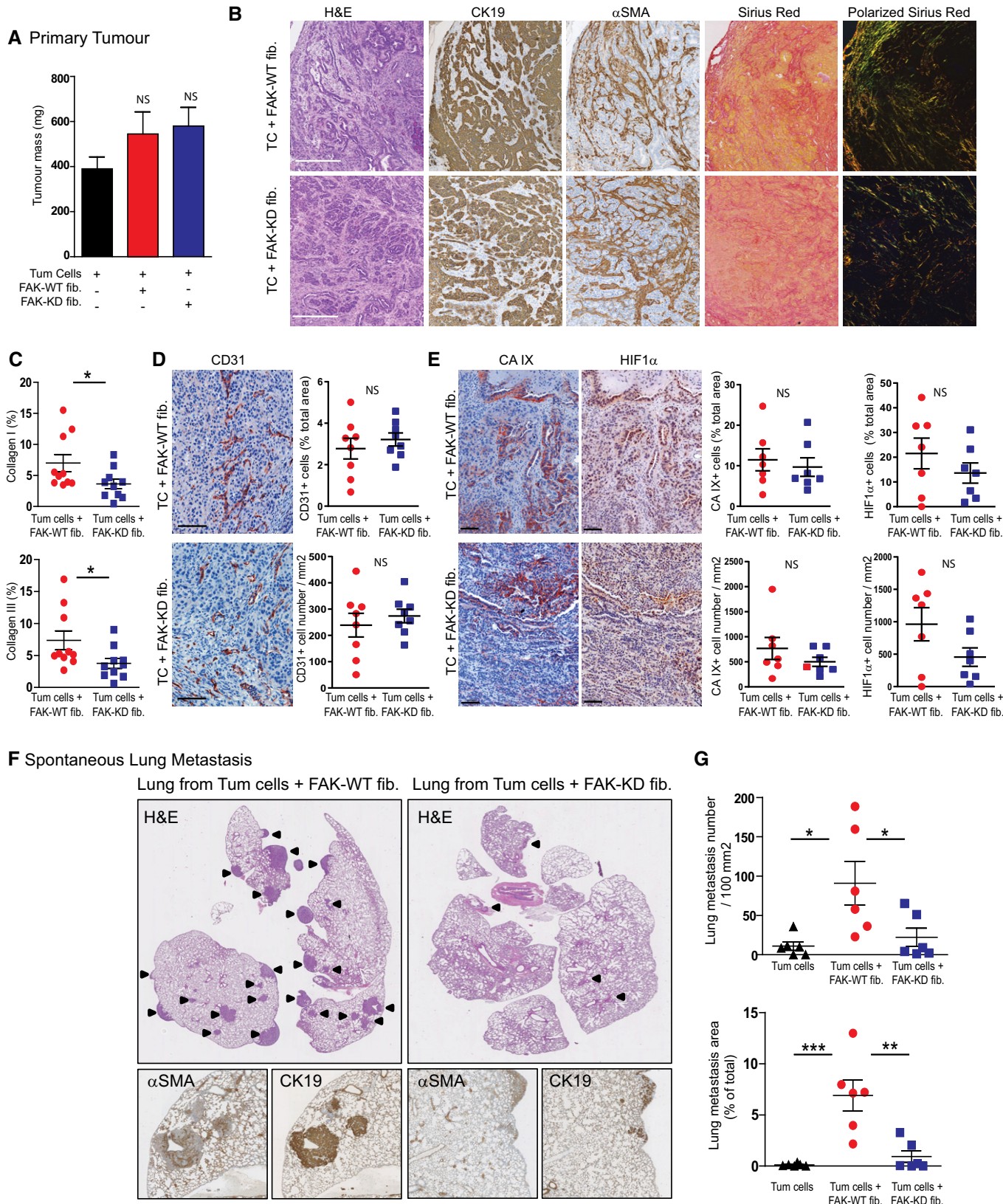

Figure 3.

**Figure 3.  Fibroblastic FAK inactivation prevents tumour metastasis without impacting tumour growth.**

Tumour cells were syngeneically and orthotopically co-grafted with mouse FAK-WT or FAK-KD fibroblasts. Thirty-eight days after injection, mice were euthanized.

A    Primary tumour mass quantification. Values are means ± SEM from at least seven mice per group. NS: not significantly different using one-way ANOVA test with Tukey's method for multiple comparison.

B    Representative images from primary tumour stained for H&E, cytokeratin 19 (CK19), α-SMA, Sirius Red and polarized Sirius Red. Scale bar, 100 μm.

C    Quantification of fibrillar collagens I (yellow-orange) and III (green) based on polarized Sirius Red images from (B). Each dot represents a mouse (*n* = 11 per group). Values are means ± SEM, *P < 0.05 using unpaired two-tailed Student's *t*-test.

D, E  IHC representative images (left) and quantification (right) of the expression of CD31 (D) and two hypoxia markers CaIX and HIF1α (E) from primary tumour; scale bar, 100 μm. Values are means ± SEM from seven mice per group. NS, not significantly different using unpaired two-tailed Student's *t*-test.

F    Representative images of H&E, CK19 and α-SMA IHC staining performed on lung sections. Arrow identified lung metastasis.

G    Quantification of lung metastasis size and number based on CK19-positive staining using NanoZoomer software. Values are means ± SEM of six mice per group. *P < 0.05, **P < 0.01, ***P < 0.001, using one-way ANOVA test with Tukey's method for multiple comparison. NS, not significantly different.

Source data are available online for this figure.

α or PDGFR-α are GFP-positive in both groups (Fig EV2C), demonstrating that most CAFs present in the tumour, at the time of sacrifice, originate from the grafted fibroblasts.

Given the fact that CAFs are major modulators of every component of the tumour, we analysed the impact of fibroblastic FAK inactivation on primary tumour. Histopathology of the primary tumours showed that fibroblastic FAK inactivation does not impact the general architecture and the proportion of tumour cells (CK19-positive) vs fibroblasts (α-SMA-positive) (Fig 3B). However, fibroblastic FAK inactivation drastically impairs tumour collagen I and III fibre production and maturation as visualized and quantified under polarized light using Sirius red coloration (Fig 3B and C). Analysis of tumour vessels showed that FAK inactivation within CAFs does not impact epithelial cell (CD31-positive) organization and number (Fig 3D) but slightly decreases hypoxia. Indeed, the quantification of the two classically used markers of hypoxia, carbonic anhydrase IX (CA IX) and hypoxia-inducible factor 1α (HIF1α), shows a slight decrease of CA IX or HIF1α-positive cell number (Fig 3E).

Finally, as fibroblastic FAK activity in human PDAC tumours is an independent prognostic factor for DFS and OS, and mortality is mostly from metastatic disease (Siegel *et al*, 2017), we assessed the possible impact of fibroblastic FAK activity on tumour metastasis. As shown in Fig 3F and G, specific fibroblastic FAK inactivation drastically diminished PDAC spontaneous lung metastasis, decreasing both metastasis number and size. Interestingly, such metastases are rich in activated fibroblasts (α-SMA-positive cells; Fig 3F).

Taken together, these *in vivo* results show that specific FAK inactivation within fibroblasts decreases fibrosis and drastically reduces spontaneous lung metastasis.

### Fibroblastic FAK inactivation reduces M2 macrophage polarization, migration and correlates with M2 macrophage number in human PDAC samples

As CAFs and ECM may impact immune cell trafficking (Hallmann *et al*, 2015; Sun *et al*, 2018a), we postulated that fibroblastic FAK inactivation could modify immune cell populations in PDAC. Characterization of tumour-infiltrating immune cells in the primary tumour was performed by flow cytometry (gating strategy in Appendix Fig S3A and B) at early (21 days after the grafting) and late (38 days) time points. At both time points, FAK inactivation

within fibroblasts significantly decreases the pro-tumoural M2 macrophage number (CD206[+] tumour-associated macrophages), while not affecting the total macrophage number, but increasing the M1 (Nos2[+]) macrophage number only in early setting (Fig 4A and B). None of the other immune cell types are modified by the fibroblastic FAK inactivation at both time points (Fig EV3A and B). IHC analyses confirm that fibroblastic FAK inactivation significantly decreases CD206[+] tumour-associated macrophage number at specific localizations, i.e. tumoural, adjacent and fibrotic areas (Fig EV3C). Interestingly, whereas fibroblastic FAK inactivation does not impact lymphocyte T CD8[+] number in the whole tumour (Fig EV3A and B and D left), it increases the number of these cytotoxic T lymphocytes specifically within the fibrotic area (but without decreasing their number in either the tumoural or adjacent areas; Fig EV3D). Then, we searched for a mechanism underlying the reduced *in vivo* M2 tumour-associated macrophages induced by fibroblast-specific FAK inactivation. To do so, we explored the *in vitro* polarization of murine BMDM-derived M0 macrophages into M1 or M2 macrophages, upon 24-h exposure to conditioned medium (CM) collected from FAK-WT or FAK-KD activated fibroblasts (Fig 4C). Fibroblasts were first activated using CM secreted by tumour cells, and their activation was confirmed by expression increase of PDGFR-α, FAP-α and α-SMA (Fig EV3E) markers. We observed that CM from FAK-KD activated fibroblasts decreases M2 polarization (decreased percentage of CD206[high]/CMH2[low] but increased of CD206[low]/CMH2[high] cells, and decreased dectin[+] cells), without impacting M1 polarization, when compared to effect induced by CM from FAK-WT activated fibroblasts (Fig 4D). Then, we explored the impact of fibroblastic FAK pharmacological inactivation on CM-induced M1 or M2 macrophage migration, using a transwell assay. To do so, resting macrophages (M0) were first polarized into M1 or M2 macrophages by exposure to IFNγ + LPS- or IL-4 + Il-13, respectively (polarization validation in Fig EV3F). In parallel, four hCAFs (isolated from fresh patient PDAC tumours summarized in Table EV2) were treated with the FAK inhibitor PF-562271 (a pharmacological inhibitor of FAK activity), and their CM were collected. M1 or M2 macrophages were then seeded on the top chamber of the transwell and hCAF CM on the bottom chamber. We observed that both M1 and M2 macrophages migrate through the transwell between 24-h and 48-h exposure to hCAF CM and that FAK inactivation within hCAFs alters the chemoattractant potential of their secretions onto M2, but not M1, macrophages (Fig 4E). We excluded that FAK inhibitor (FAK-I) directly

impacts M1 and M2 macrophage migration as FAK-I pre-incubated for 48 h in un-conditioned CAF medium (DMEM/F12 + 0.5% foetal bovine serum [FBS] without CAF) does not change macrophage migration (Fig EV3G). These data demonstrate that FAK activity within CAFs positively regulates the secretion of soluble factors that polarize macrophages towards the M2 phenotype and

enhances their migration. Therefore, we searched for the involved cytokines/chemokines.

Tumour-associated macrophages originate from circulating monocytes, which are recruited to the tumour by several growth factors and especially by chemokines, produced by stromal and tumour cells (Lahmar *et al*, 2016). Besides M-CSF, the CC

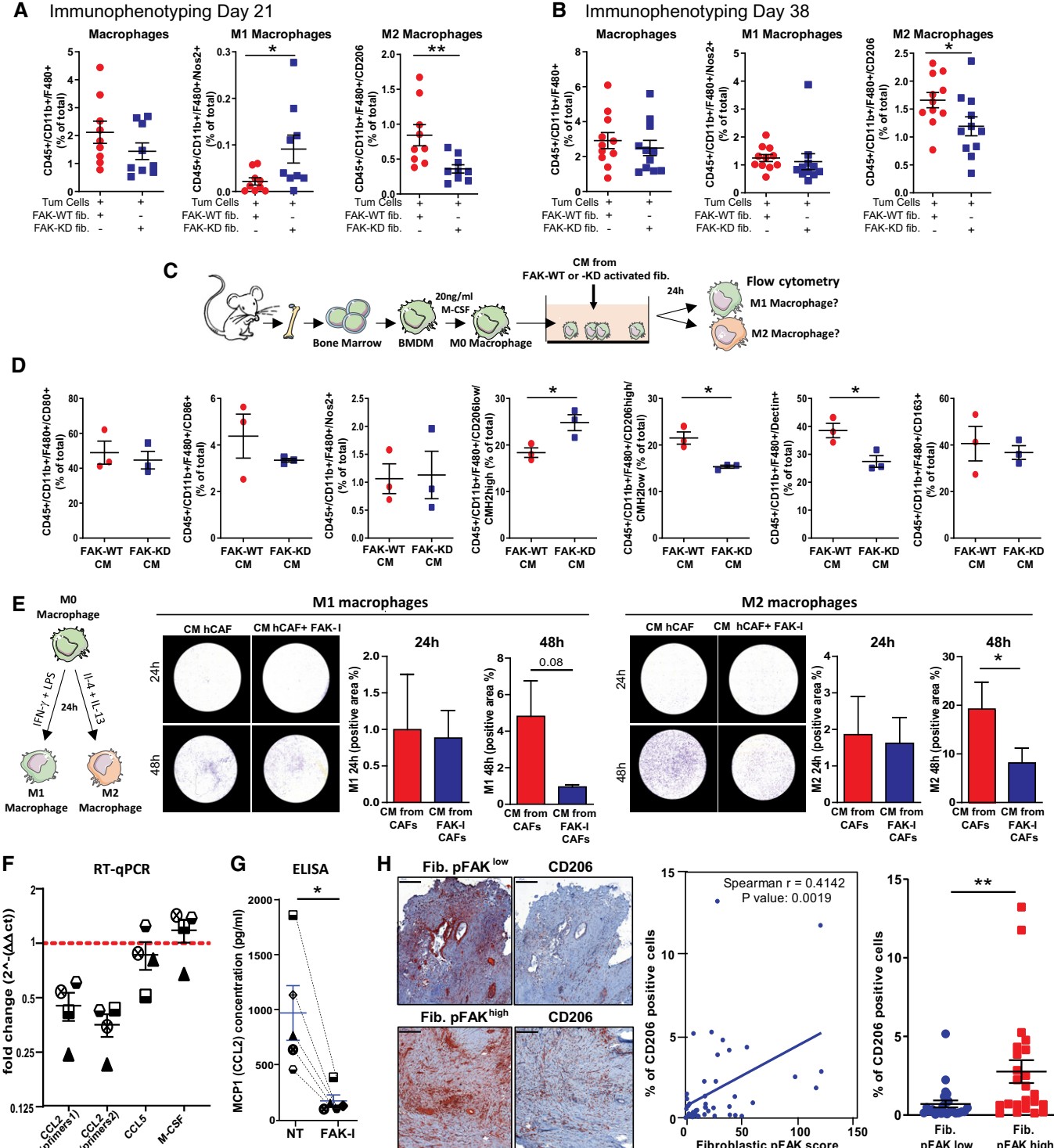

**Figure 4.**

**Figure 4.   Fibroblastic FAK inactivation reduces M2 macrophage polarization, migration and correlates with M2 macrophage number in human PDAC samples.**

A, B   Relative frequencies of tumour-infiltrating M1 macrophages and M2 macrophages analysed by flow cytometry at 21 days (A) and 38 days (B) after grafting. Values are means ± SEM from 5 to 10 mice per group, *$P < 0.05$, **$P < 0.01$ using unpaired two-tailed Student's *t*-test.

C   Schematic of BMDM isolation and macrophage polarization assay.

D   Impact of conditioned media (CM) secreted by activated mouse fibroblasts expressing FAK-WT or FAK-KD on mouse BMDM-derived macrophage polarization. Relative frequencies of macrophages expressing CD80, CD86, Nos2, CD206, CMH2, dectin and CD163 markers acquired by flow cytometry and processed using FlowJo software. *$P < 0.05$, by unpaired two-tailed Student's *t*-test; values are mean ± SEM.

E   Impact of soluble factors secreted by CAFs on macrophage migration. Left: schematic of BMDM-derived M1 or M2-like macrophages. Right: Representative pictures and quantification of M1 and M2 macrophage migration at 24 and 48 h towards CM secreted by CAFs (previously treated or not with FAK inhibitor for 48 h). Cells were stained using crystal violet and percentage of crystal violet positive area quantified using Cell Observer videomicroscope (Zeiss) and analysed with ImageJ. *$P < 0.05$, by paired two-tailed Student's *t*-test; values are mean ± SEM from three experiments.

F   Quantitative real-time PCR conducted with gene-specific primers (CCL2, CCL5 and M-CSF) on four different FAK-I-treated (16 h) and non-treated CAFs for gene expression. The normalized fold change ± SEM was calculated using the delta-delta Ct method with RPS16 as a control gene; values are mean ± SEM. Each individual patient-derived activated fibroblast was assigned to a specific symbol (see Table EV2).

G   ELISA quantification of CCL2 (MCP1) concentration (µg/ml) in conditioned medium from hCAFs treated or not with FAK-I during 72 h. *$P < 0.05$, by paired two-tailed Student's *t*-test values are mean ± SEM. Each individual patient-derived activated fibroblast was assigned to a specific symbol (see Table EV2).

H   Left: Representative immunohistochemistry pictures of phosphorylated FAK on tyrosine (Y) 397 and of CD206 in serial sections of human PDAC tissues. Scale bar: 200 µm. Middle: Correlation plot between fibroblastic pY397 FAK score and CD45 or CD206⁺ cell number within human PDAC samples. Fibroblastic pY397 FAK score and percentage of CD206-positive cell quantifications were performed on a 47 patient cohort based on IHC staining. Spearman $r = 0.4145$, $P = 0.0019$. Right: Percentage of CD206-positive cells in PDAC patient expressing either low or high fibroblastic FAK activity score. Values (blue bars) are means ± SEM; **$P < 0.01$ using unpaired two-tailed Student's *t*-test.

Source data are available online for this figure.

chemokines CCL2, CCL3, CCL4 and CCL5 are well-recognized chemotactic factors for macrophage populations in the tumour (Ruytinx *et al*, 2018). mRNA expression levels of M-CSF, CCL2, CCL3, CCL4 and CCL5 were analysed in four hCAFs treated with FAK-I. CCL3 and 4 mRNA were undetectable. M-CSF and CCL5 mRNA expressions are not altered by FAK-I treatment; however, CCL2 is drastically reduced at both mRNA (59.4% decrease; Fig 4F) and protein level (84.9% decrease; evaluated by ELISA in hCAF CM) compared to untreated hCAFs (Fig 4G).

Finally, to validate that fibroblastic FAK activity influences M2 macrophages in patients, we searched for a correlation between CD206⁺ tumour-associated macrophage abundance and the fibroblastic pY397 score on a cohort of 47 PDAC patients. As shown in Fig 4H, FAK activity within CAFs positively correlates with CD206⁺ macrophage number ($P = 0.0019$; $r = 0.4142$).

Taken together, we conclude from this section that fibroblastic FAK inactivation decreases intra-tumour M2 macrophage presence. FAK activity within CAFs regulates CCL2 mRNA and secreted protein expression, favours *in vitro* M2 macrophage polarization and migration, and positively correlates with CD206⁺ macrophage number within human PDAC tumours.

### Fibroblastic FAK activity controls tumour cell migration and invasion

We then undertook to understand how the sole inactivation of fibroblastic FAK within the primary tumour dramatically reduces spontaneous metastasis, and hypothesized a role for CAF-induced cancer cell invasiveness. Migration of green-labelled FAK-WT or FAK-KD fibroblasts co-cultured with red-labelled KPC cancer cells was explored in a 2D scratch wound assay. Videomicroscopy shows that FAK inactivation in fibroblasts delays the wound closure time from 46 h to more than 72 h (Movies EV1 and EV2 and Fig 5A). Three major parameters were analysed and quantified based on cell tracking (Fig 5B–H, Movies EV1 and EV2): cell velocity (rapidity of cell motion, calculated on moving cells), directionality and distance of migration (length of the path travelled). We observed that, in co-

cultures (fibroblasts plus tumour cells), fibroblasts migrate first, independently of whether they express FAK-WT or FAK-KD, and are followed by tumour cells (Movies EV1 and EV2). When fibroblasts are cultured alone, FAK inactivation significantly decreases their velocity (Figs 5C and, EV4A and B), as well as drastically impairs their migration directionality (Fig 5D and, EV4A and B), confirming published results (Lim *et al*, 2010). Adding tumour cells, which activate fibroblasts (Fig EV3E and (Apte & Wilson, 2012)), rescues FAK-KD fibroblast velocity to levels observed with FAK-WT fibroblasts co-cultured with tumour cells (Fig 5C), but not their loss of directionality (Fig 5D). Moreover, whereas migration distances of FAK-WT and FAK-KD fibroblasts are comparable when cultured alone (Fig 5E), the presence of tumour cells robustly increases the migration distance of activated FAK-KD, but not of FAK-WT, fibroblasts. Indeed, FAK-KD fibroblasts migrate randomly (with low directionality, Movie EV2) and therefore the wound does not close, resulting in continued migration of fibroblasts, which explains their increased migration distance.

Tumour cell velocity is not impacted by the presence of either FAK-WT or FAK-KD fibroblasts (Fig 5F). However, tumour cells migrate with a more linear manner (increased directionality) when co-cultured with FAK-WT fibroblasts compared to tumour cells alone, but not with FAK-KD fibroblasts (Fig 5G). As a result, the distance travelled by migrating tumour cells is significantly higher when co-cultured with FAK-WT, but not with FAK-KD, fibroblasts (Fig 5H). These results show that fibroblastic FAK activity governs, at least in part, the directionality and the subsequent distance travelled by migrating tumour cells.

Next we explored how fibroblastic FAK is involved in cancer cell invasion using a 3D co-culture assay. Visualization and quantification of cell invasion was assessed on spheroids, composed of pancreatic tumour cells and FAK-WT or FAK-KD fibroblasts, embedded into a collagen I matrix (Fig EV4C). Figure 5I shows that fibroblasts expressing FAK-WT are the first cells to invade the collagen matrix and are followed by tumour cells. These sequential invasion events mimic what happens in our *in vivo* model, whereby fibroblasts (α-SMA-positive cells) precede tumour cells (cytokeratin 19-

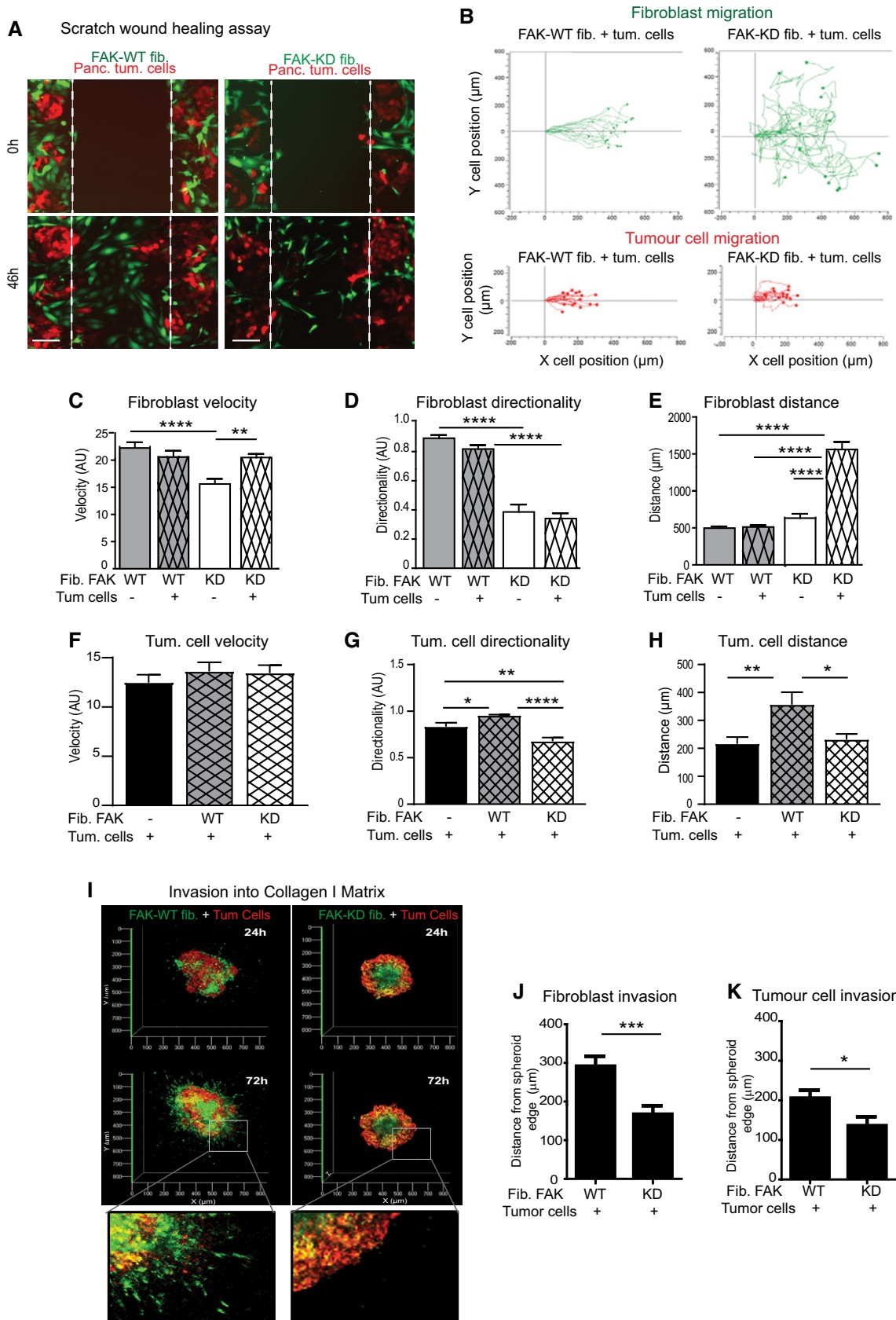

Figure 5.

**Figure 5. Fibroblastic FAK activity controls tumour cell migration and invasion.**

A Representative images of three independent scratch wound assays of co-cultivated red-labelled pancreatic tumour cells with either FAK-WT (wild-type) or FAK-KD (kinase-dead) green-labelled fibroblasts at time zero and 46 h (corresponding to the wound closure of co-cultured FAK-WT fibroblasts with tumour cells). Scale bar, 150 μm.

B–H Cell migration has been analysed in detail by tracking the location of cells (*n* > 16 cells per condition) over time for fibroblasts or tumour cells (B), and measuring fibroblast velocity (C), directionality (a cell with a directionality equal to 1 is moving straightforward) (D) and migration distance (E) or tumour cell velocity (F), directionality (G) and distance (H). Values are means ± SEM of 16 tracked cells from two independent experiments. *$P < 0.05$, **$P < 0.01$, ***$P < 0.001$, ****$P < 0.0001$ by one-way ANOVA with Tukey's method for multiple comparison.

I 3D reconstitution of cell invasion into collagen matrix from spheroid composed of green-labelled FAK-WT or FAK-KD fibroblasts plus red-labelled pancreatic tumour cells (ratio 1/1). Shown is representative 3D reconstitution of spheroid invasion into 2 mg/ml collagen I matrix obtained using Zen software.

J, K Quantification of fibroblast (J) or tumour cell (K) invasion within collagen matrix. Values are means ± SEM of 60 tracked cells from three independent experiments. *$P < 0.05$, ***$P < 0.001$, by unpaired two-tailed Student's *t*-test.

Source data are available online for this figure.

positive cells) during the invasion process into the spleen (Fig EV4D). Importantly, loss of FAK activity specifically within fibroblasts drastically modifies spheroid organization and cell invasion (Fig 5I), whereby distances of fibroblast (Fig 5J) and of pancreatic tumour cell (Fig 5K) invasion from the spheroid edge are significantly decreased.

Our results show that fibroblast-induced tumour cell migration and invasion are dependent on fibroblastic FAK activity.

**Fibroblastic FAK activity promotes drastic ECM modification**

We then undertook to identify mechanisms implicated in the fibroblastic FAK-induced tumour cell invasion and postulated that ECM remodelling could be involved. Indeed, in collective migration, carcinoma cells move along invasive tracks made of ECM deposited by fibroblasts (Beacham & Cukierman, 2005; Gaggioli *et al*, 2007) and mechanically remodelled by them (Cukierman *et al*, 2002; Grinnell, 2003). ECM tracks play important role in the initial metastatic step (Kopanska *et al*, 2016). Thus, we analysed for changes in ECM, cancer cell and/or CAF organization at the invasive front in our *in vivo* model and identified a drastic difference between the two groups. Figure 6A shows that, in the FAK-WT fibroblast tumours, CAFs are oriented towards the adjacent pancreatic tissue, thus with a radial arrangement (parallel to the tumour radius but aligned perpendicularly to the tumour edge), but FAK-KD CAFs are organized perpendicular to the tumour radius (parallel to the tumour edge). Interestingly, ECM (Sirius Red) and CAFs have the same orientation (Figs 6A and EV5A). We then undertook to characterize, *in vitro*, the impact of FAK inactivation on deposited ECM during fibroblast migration. First of all, FAK-WT and FAK-KD fibroblasts were activated by TGF-β treatment as this cytokine is known to promote fibroblast

**Figure 6. Fibroblastic FAK activity promotes drastic ECM modification.**

A Representative images of CK19 and α-SMA immunohistochemistry and Sirius red staining in serial sections of primary mouse tumour (graft of FAK-WT or FAK-KD fibroblasts + tumours cells at 38 days after injection) at the invasive front. Scale bar, 100 μm. Bottom: schematic organization of the cells (tumour cells in brown, CAFs in yellow and non-transformed epithelial cells in beige).

B Representative images of ECM deposition (stained using Alexa Fluor 488 succinimidyl ester [NHS]) generated by FAK-WT or FAK-KD activated fibroblasts during migration. Scale bar, 50 μm.

C Quantification of collagen I gel contraction (calculated based on photographed gels using the formula 100*(well diameter − gel diameter)/well diameter) induced by fibroblasts in presence or not of tumour cells (left: one-way ANOVA with Tukey's method for multiple comparison), or by hCAFs in presence or not of FAK inhibitor (FAK-I, 1 μM; right: paired two-tailed Student's *t*-test for each time). Values are means ± SEM from three independent experiments of triplicates. *$P < 0.05$, **$P < 0.01$, ***$P < 0.001$.

D Quantification of the percentage change of Y397 FAK, total FAK, collagen I (coll I), collagen III (coll III), collagen IV (coll IV), LOXL2, periostin (POSTN), osteopontin (OPN) expression by human CAFs treated or not (NT) with FAK inhibitor (1 μM), based on immunofluorescence analyses. Values are means ± SEM obtained from at least five primary hCAFs, isolated from 5 to 12 different fresh human PDAC samples. NS: not significantly different; *$P < 0.05$, **$P < 0.01$, ***$P < 0.001$, ****$P < 0.0001$ using paired two-tailed Student's *t*-test.

E Collagens I, III and IV and periostin deposition are shown using heat map of the IF staining, and cell shapes are identified by white line. Scale bar, 50 μm.

F–K Characterization of the ECM-enriched fraction (matrisome analyses) from hCAFs treated or not with FAK-I by LC-MS/MS proteomics. (F) Pie charts display the relative proportion of each of the major classes of matrisome proteins in non-treated hCAFs (*N* = 3 biological replicates with two technical replicates/condition) vs FAK-I-treated hCAFs (*N* = 3 biological replicates with two technical replicates/condition). (G, H) Left: Normalized peptide abundance for collagens (G) or proteoglycans (H) from hCAFs (each CAF duplicate is represented, CAF1 in blue, two in red and three in black) treated or not with FAK inhibitor. Right: Heatmap of the collagen (G) or proteoglycan (H) modifications in the three hCAFs upon FAK inhibition in hCAFs. The value of log$_2$ fold change between FAK inhibitor-treated CAFs and non-treated CAFs for each protein was indicated by the coloured scale, with red indicating increased expression while blue implies decreased expression. For each CAFs and each protein, log$_2$ fold change was calculated based on two technical replicates per condition (biological replicate *n* = 3, technical replicate *n* = 2). (I) Normalized peptide abundance for glycoproteins from hCAFs treated or not with FAK-I. (J) Heatmap of the glycoprotein modifications upon FAK-I in hCAFs. (K) Normalized peptide abundance for "matrisome-associated" protein division: ECM-affiliated proteins, secreted factors and ECM regulators from hCAFs treated or not with FAK inhibitor. (G, H, K) Each individual patient-derived activated fibroblast was assigned to a specific symbol (see Table EV2).

L Left: Representative pictures of β1 integrin activation (red) of tumour cells, upon adhesion to either ECM deposited from non-treated CAFs or treated with FAK-I for 7 days, analysed by IF. Merge: activated β1 integrin (red), phalloidin (green), dapi (blue). Right: quantification of the number of activated integrin β1 cluster per cells and size of those clusters. Values are means ± SEM, ****$P < 0.0001$ using paired two-tailed Student's *t*-test on at least nine images (of three to 33 cells per image) per group, scale bar: 10 μm.

Source data are available online for this figure.

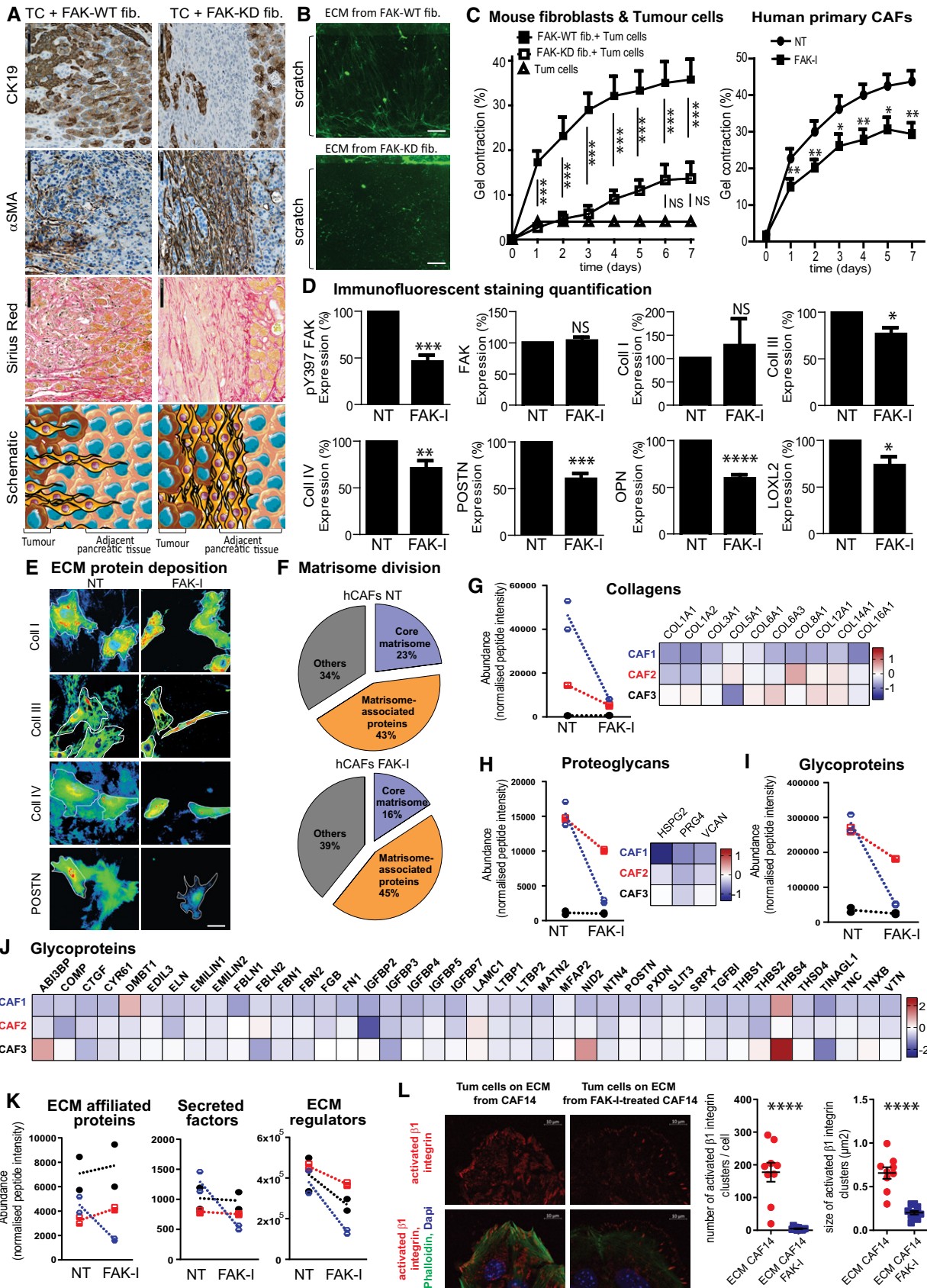

**Figure 6.**

activation during wound healing and tumorigenesis (Desmouliere *et al*, 1993; Phan, 2008; Albrengues *et al*, 2014). Analysis of deposited ECM during fibroblast migration shows that FAK activity is required for the production of ECM tracks as TGF-β-activated FAK-WT fibroblasts generate dense and abundant ECM fibres, oriented through the wound, whereas FAK-KD fibroblasts instead deposit ECM in heaps (Fig 6B). Thus, FAK activity within fibroblasts regulates ECM track generation both *in vitro* and *in vivo*.

Extracellular matrix track generation is dependent on fibroblast-induced ECM remodelling. ECM remodelling occurs through forces applied by cells on the ECM through integrins (named force-mediated matrix remodelling), and can be quantified by measuring collagen matrix contraction (Cukierman *et al*, 2002; Grinnell, 2003). Therefore, we quantified the impact of FAK inhibition on fibroblast-induced ECM contraction. In basal conditions, FAK-WT fibroblasts display a higher contractile phenotype than FAK-KD fibroblasts (Fig EV5B–D). A transient pulse of TGF-β stimulation increases force-mediated matrix remodelling in FAK-WT, but not in FAK-KD, fibroblasts (Fig EV5B–D). Similarly, fibroblasts activated by the presence of tumour cells are less susceptible to contract the matrix when their FAK is inactivated (Fig 6C). Importantly, pharmacological FAK inactivation in primary hCAFs also significantly decreases CAF-induced matrix contraction (Fig 6C). Thus, FAK activity within fibroblasts regulates ECM organization by impacting force-mediated matrix remodelling.

Extracellular matrix composition and organization are key regulators of the initial step of metastasis. Thus, we analysed the impact of FAK inactivation on CAF-derived ECM production and composition using primary cultures of hCAFs. As expected, FAK inhibition decreases FAK-Y397 phosphorylation without impacting total FAK expression (Figs 6D and EV5E–G). FAK inactivation does not significantly impact collagen I expression (although 40% of hCAF culture show a decrease), but a significant downregulation is seen for collagen III and IV expression (Figs 6D and EV5E–G) and extracellular deposition (Fig 6E), as well as for LOXL2 expression, an enzyme implicated in the last step of collagen fibre maturation and more generally in matrix remodelling (Barker *et al*, 2013) (Figs 6D and, EV5E and F). Upon FAK inactivation, 100% of the hCAF culture show a decrease in periostin expression and deposition (Fig 6D and E), as well as in osteopontin expression (Figs 6D and EV5E–G), two matricellular proteins implicated in tumour progression (Kolb *et al*, 2005; Fukushima *et al*, 2008).

To better identify changes in the composition of ECM protein expression induced by fibroblastic FAK inactivation, mass spectrometry-based proteomics, a method of choice to profile the protein composition of ECMs produced by cells in culture, was performed (data are available via ProteomeXchange with identifier PXD018899). Using the matrisome-focused proteomic technique (Naba *et al*, 2017), we quantitatively assessed matrisome proteins on 3 primary cultures of hCAFs treated for 7 days with FAK inhibitor (1 μM). This analysis shows that FAK inhibition robustly decreases the expression of proteins belonging to the "core matrisome" (structural and fibrillar extracellular components, glycoproteins and proteoglycans), which drops from 23 to 16% of the whole matrisome content. In contrast, "matrisome-associated proteins" (ECM regulators, ECM-affiliated proteins and ECM-secreted factors) are globally not impacted by FAK inhibition (Fig 6F). Figure 6G–J shows that, among the three hCAF cultures tested, two (hCAF1 and

2) present, under FAK inactivation, a strong decrease of collagens (10 different collagens were detected, Fig 6G), proteoglycans (three detected, Fig 6H) and glycoproteins (40 proteins detected, Fig 6I and J). Importantly, as opposed to hCAF1 and 2, basal levels of collagens, proteoglycans and glycoproteins produced by hCAF3 are very low and are not impacted by FAK inhibition (Fig 6G–I). Such differences in terms of response to treatment are in accordance with our immunofluorescence (IF) results (Fig EV5E and F) and were expected as recent publications have reported hCAF heterogeneity in PDAC tumours (Ohlund *et al*, 2017). Regarding "matrisome-associated proteins" that are globally not modified by FAK inhibition (Fig 6K), ECM regulators (32 detected) are downregulated in all three hCAFs, whereas ECM-affiliated proteins (affiliated either structurally or physically with the core matrisome, 14 detected) and secreted factors (such as growth factors and cytokines, known or suspected to bind to ECM, 17 detected) show no change in expression levels (Figs 6K and EV5H–J).

Finally, we analysed the impact of such ECM modifications on tumour cell integrin activation and more specifically on β1 integrin, as overexpression of β1 is correlated with poorer DFS in PDAC (Sun *et al*, 2018b), and β1 integrin activation mediates adhesion and invasiveness in PDAC lines (Arao *et al*, 2000). We report here that tumour cell β1 integrin activation is dramatically decreased upon adhesion to ECM deposited by hCAFs treated with FAK-I compared to non-treated hCAFs, while β1 integrin expression is not changed (Figs 6L, EV5K and L).

Altogether, these data show that pharmacological FAK inactivation prevents hCAF force-mediated matrix remodelling and down-regulates expression of core matrisome proteins. FAK inhibition reduces hCAF-induced collagen, glycoprotein, proteoglycan and ECM-regulator expression without impacting ECM-affiliated proteins and secreted factors. Such modifications directly influence tumour cell integrin activation status.

## Discussion

In this study, we report that FAK activity within CAFs is an independent prognostic marker for PDAC patients (patients with high FAK activity, i.e. high score for pY397 FAK within CAFs, have significantly shorter DFS and OS than patients with low FAK activity), whereas FAK activity within tumour cells is not. These cell specific observations are in accordance with a recent publication by Jiang *et al* (2016) that reports that the tumour cell pY397 FAK level does not correlate by itself with patient survival, but only when combined with intra-tumoural CD8[+] cytotoxic T lymphocyte infiltration.

When splitting our cohort into high grade ($n = 22$) and low or medium grade ($n = 98$) tumour patients, it appears that fibroblastic FAK activity is predictive of shorter DFS or OS for the less aggressive group (low and moderate grade tumours). We suggest that a gain of FAK activity within CAFs may provide an invasive advantage to well and differentiated tumour cells endowed with a low intrinsic invasive capacity (no epithelial–mesenchymal transition, EMT), and for which the biology of the tumour invasion process is still not clear. Instead, in aggressive tumours, dedifferentiated carcinoma cells (high grade), present at invasive tumour margins, detach from the main tumour, (a process named "tumour budding"

(Karamitopoulou *et al*, 2018)), and acquire EMT features, so intrinsic skills for invasion. Thus, we suggest that CAFs with high FAK activity do not change tumour cell intrinsic features for invasion but provide these features extrinsically to lower grade tumour cells. This may explain why fibroblastic FAK activity does not correlate with tumour grade, even so it is a well-established prognostic factor in PDAC.

Of note, tumour size, grade, lymph node status and resection margins are the main tumour-derived prognostic factors in resected PDAC patients. However, they are insufficient to fully predict the risk of recurrence and death in resected PDAC patients. Thus, there is a need for biomarkers to improve PDAC patient prognostic stratification. Our results indicate that fibroblastic FAK activity is a prognostic marker independent of all other known "classical" predictive markers (i.e. grade, tumour size, lymph nodes, resection margins, perineural and vascular invasion), and support that fibroblastic FAK activity might improve the prognostic stratification of the "differentiated-grade" subgroup of patients for which actual histopathological markers are far to be perfect.

To our knowledge, the status and role of FAK activity within CAFs were unknown. FAK signalling and role have mainly been studied in "normal" fibroblasts where it regulates cell adhesion, spreading, migration and survival (Sieg *et al*, 2000; Schlaepfer & Mitra, 2004; Horowitz *et al*, 2007; Lim *et al*, 2010), as well as profibrotic gene expression and matrix remodelling (Ilic *et al*, 2004; Hong *et al*, 2010). Additionally, in pathologic conditions, pharmacologic FAK inactivation was reported to reduce the number of CAFs within PDAC primary tumours or to abrogate lung fibrosis on bleomycin-induced fibrosis mouse model, but without any demonstration that targeting fibroblastic FAK activity may be involved (Stokes *et al*, 2011; Lagares *et al*, 2012; Kinoshita *et al*, 2013). Here we show that specific FAK inactivation in CAFs (using models of pharmacologic or genetic inactivation) impairs fibroblast and tumour cell migration and invasion *in vitro* and reduces metastasis formation *in vivo*. Moreover, FAK activity within fibroblasts is critical for fibroblast-induced ECM remodelling. Indeed, using the "matrisome" approach developed by R. Hynes and A. Naba (http://matrisomeproject.mit.edu), we show that pharmacologic FAK inhibition in primary CAFs isolated from patient tumours globally decreases the expression of ECM proteins from the "core matrisome" (collagens, glycoproteins and proteoglycans) and ECM regulators. Using more classical technics (i.e. IF and Western blot), we validated, *in vivo* and *in vitro*, that FAK inactivation decreases the expression and deposition of key ECM proteins (collagens III and IV, periostin and osteopontin) and of LOXL2, an enzyme involved in collagen fibre maturation. This is consistent with our observation that tumour collagen bundling is impaired in mouse FAK-KD co-engrafted tumours. Altogether, it appears that the prognostic value of FAK activity within CAFs in PDAC patients could be linked to the stimulating effects of FAK signalling on CAF-induced ECM production and remodelling, two critical triggers of tumour cell metastasis (Provenzano *et al*, 2006; Erdogan & Webb, 2017). Interestingly, Moffitt *et al* (2015) recently established that PDAC patients with an "activated stromal" subtype, which comprises enrichment in ECM genes (collagens, fibronectin and periostin), have reduced survival as compared to patients with a "normal" stromal subtype. In 2018, Puleo *et al* (2018), based on a larger cohort of 309 PDAC samples, redefined the classification allowing an integrated stratification using

tumoural and stromal compartments. An activated stroma component defined by high FAP, and ACTA2 expression, correlates with the activated stromal signature defined by Moffitt *et al*. Enrichment analysis confirmed significant overexpression of activated stroma, extracellular matrix organization, focal adhesion, extracellular matrix–receptor interaction and collagen formation pathway genes (Puleo *et al*, 2018). Since we demonstrate that fibroblastic FAK inhibition decreases ECM production and remodelling, one can predict that targeting pancreatic CAFs using a FAK pharmacological inhibitor will induce a stroma switch from the "activated" to the "normal" phenotype.

We here report that FAK activity within CAFs correlates with patient survival, drives ECM remodelling resulting in tumour invasion and favours M2 pro-tumoural macrophage recruitment via MCP1 secretion. When analysed in detail, we noticed that human primary PDAC CAFs present an important inter-patient heterogeneity in terms of FAK activity score, of MCP1 gene and protein expression, of production and secretion of specific ECM, of effectiveness to recruit pro-tumoural macrophages, and of response to FAK inhibitor. Inter-tumour stromal heterogeneity was reported in PDAC patients and enables patient stratification within prognosis groups (Moffitt *et al*, 2015; Puleo *et al*, 2018). Additionally, diverse subpopulations of CAFs were recently reported to co-exist within a same PDAC tumour, being heterogeneous with regard to cell surface marker expression, cytokine production or cell signalling (Nielsen *et al*, 2018; Whittle & Hingorani, 2019; Norton *et al*, 2020). A recent publication identified, using molecular and functional analyses of several patient-derived CAF primary cultures, four CAF sub-groups that co-exist within a tumour, each featuring specific phenotype and prognostic value (Neuzillet *et al*, 2019). While we were preparing this manuscript, the Hodivala-Dilke's group showed that FAK depletion in a subpopulation of CAFs (FSP1-positive) metabolically increases pancreatic tumour growth in mice and that low FAK gene expression within CAFs is associated with poor PDAC patient survival (Demircioglu *et al*, 2020). FAK role depends on both its scaffolding and kinase activities, and our FAK -inactivating strategy mimics the pharmacological inhibition as FAK-KD mutant retains the cytoplasmic and nuclear scaffolding functions of FAK, which is not the case when FAK gene is invalidated (Demircioglu *et al*, 2020). Another explanation for differences between ours and published results (Demircioglu *et al*, 2020) is that FAK expression and/or signalling may be heterogeneous across patient tumours and within different CAF subsets in a same tumour, putatively explaining different biological consequences for FAK targeting. Accordingly, we observed that fibroblastic FAK activity is highly heterogeneous between PDAC patients, and showed that the fibroblastic pY397 FAK score is an independent prognosis marker that would help to identify patients with high risk of early relapse after surgery, and with the most benefit of receiving an adjuvant protocol including a FAK inhibitor.

Most PDAC patients die from a metastatic disease within the first year of diagnosis. Here, we report that fibroblastic FAK inactivation drastically reduces spontaneous metastasis *in vivo*. Dissemination of cancer cells into normal adjacent tissue is driven by ECM remodelling induced either by CAFs or tumour cells themselves (Gaggioli *et al*, 2007; Friedl & Wolf, 2008). Invasive leading cells protrude, attach to the ECM and generate anterior traction forces, whereas ECM fibres are cleaved several micrometres posterior to the leading

edge (Wolf *et al*, 2007; Friedl & Wolf, 2008). Remaining matrix fibres bound to integrins are realigned as the cell moves forward, resulting in the generation of paralleled fibres, also named as invasive matrix tracks, that "follower" cells can use to migrate (Wolf *et al*, 2007; Friedl & Wolf, 2008; Kopanska *et al*, 2016). Here, we show that CAFs represent "leader" cells for the migration and invasion of pancreatic tumour "follower" cells (Fig 6). Those results can be compared to our *in vivo* observation where CAFs precede tumour cells at tumour invasive front and during spleen invasion. When fibroblastic FAK is active, CAFs invade and reorganize the ECM within adjacent tissue or distant invaded organ, giving an ECM orientation perpendicular to the tumour edge. Importantly, CAFs and ECM orientation are drastically modified when fibroblasts express an inactivated form of FAK, both presenting an orientation parallel to the tumour edge (Fig 6). ECM geometrical and mechanical properties play an important role for the initial step of metastasis formation (Provenzano *et al*, 2006). Aspects of stromal biology including reorganization of the ECM to promote invasion and changes in the expression of stromal cell types are key medical markers for cancer prognosis (Conklin & Keely, 2012). We show that FAK inactivation totally blocks the ability of CAFs to generate ECM tracks that tumour cells use to invade. Additionally, we demonstrate that FAK activation within CAF "leader" cells drives tumour "follower" cell migration and invasion since the sole inhibition of fibroblastic FAK severely decreases fibroblast and tumour cell migration and invasion *in vitro,* as well as severely reduces spontaneous metastasis *in vivo.* Altogether, the anti-metastatic effect of fibroblastic FAK inhibition appears to be linked to reduced fibroblast migration and invasive abilities, and to ECM remodelling leading to a decrease of integrin activation in tumour cells. Accordingly, high fibroblastic pY397 FAK is predictive of worse DFS and OS for the low and moderate grade group (in which tumour cells are still devoid of intrinsic invasive capacity) but not for the high grade group (in which tumour cells are able to perform EMT).

We also show that FAK activity in CAFs positively correlates with the quantity of immunosuppressive M2 macrophages in human PDAC samples. Previous studies reported that pharmacologic FAK inhibition, on genetic PDAC mouse (KPC) (Jiang *et al*, 2016) or immunocompetent PDAC-bearing mouse models (Stokes *et al*, 2011), decreased numbers of tumour-infiltrating immunosuppressive cells such as tumour-infiltrating myeloid-derived suppressor cells (MDSCs), CD206[+] tumour-associated macrophages and CD4[+]FOXP3[+]regulatory T cells (Tregs). Complementary to global pharmacologic FAK inhibition's studies, here, we report that inactivation of FAK specifically in fibroblasts impacts pro-tumoural M2 macrophage population, decreasing their migration and their polarization via a mechanism involving CAF-induced secreted factors, especially CCL2 (MCP1). The role of FAK in the regulation of secreted factors was recently reported by Serrels *et al* (2015). The authors identified that, in squamous cell carcinoma, nuclear translocation of FAK triggers transcription of inflammatory cytokines and chemokines by tumour cells, e.g. MCP1 known to facilitate M2 macrophage polarization and migration, thereby promoting an immunosuppressive microenvironment. As we show that FAK activity is increased in both the cytoplasm and nucleus of CAFs in PDAC samples (Fig 1), one can speculate that such mechanism may be involved in the FAK-dependent MCP1 secretion that we showed in CAFs.

Our results, altogether with DeNardo's group study (Jiang *et al*, 2016), provide a scientific rationale to support the ongoing clinical trials (NCT02428270, NCT02546531, NCT02651727 and NCT02758587) for PDAC patients, combining the well-tolerated FAK inhibitors (defactinib or VS-4718 or GSK2256098), with MEK inhibitor (trametinib), or pembrolizumab (a monoclonal anti-PD1 antibody) or chemotherapy (gemcitabine ± nab-paclitaxel) (Infante *et al*, 2012). FAK pharmacological inhibition should, on one hand, target tumour cells leading to increased sensitivity to chemotherapy and immune-checkpoint inhibitor (Jiang *et al*, 2016), and, on another hand, impact activated fibroblasts reducing ECM remodelling and PDAC metastasis as we demonstrate here.

# Materials and Methods

### Tissue array

#### *Commercial tissue microarray*
Commercial tissue microarray was obtained from BIOMAX.US (http://www.biomax.us/tissue-arrays/Pancreas/PA1002a) and IHC was performed as described in the "Immunostaining" section. Slides stained for pY397 FAK were imaged using NanoZoomer XR Hamamatsu (Imag'IN core from Institut Universitaire du Cancer Toulouse, France). Two spots per patient were double blinded quantified based on pY397 FAK staining intensity and positive cell number (a total of four quantifications were performed per patient, and mean calculated). pY397 FAK cytoplasmic and nuclear staining were both scored on a scale of 6: 0 for no staining and 6 for high positive staining, taking into account the intensity of the staining and the quantity of positive cells. pY397 FAK CAF staining score is the mean of cytoplasmic and nuclear score.

### Patient samples

#### *120 patient cohort, tissue array*
Patients were selected from a retrospective cohort of 120 consecutive patients with stage I–II PDAC who underwent complete surgical resection between April 1997 and April 2009 at Beaujon University Hospital (Clichy, France). All participants gave informed consent before taking part in accordance with the principles set out in the WMA Declaration of Helsinki and the Department of Health and Human Services Belmont Report. Exclusion criteria were macroscopically incomplete resection (R2), tumour histology other than PDAC and death within 30 days following surgery. TMAs were constructed from 600 μm tissue cores obtained from formalin-fixed paraffin-embedded tumour samples, with 3–5 cores per patient, and used for prognostic and predictive biomarker research. Clinical and pathological variables were retrospectively collected, including sex, age at diagnosis, tumour site, preoperative assessment of tumour extension, surgical procedure, resection margins (R0: negative, R1: positive), differentiation grade, tumour size, tumour stage, lymph node status (negative: N−, positive: N+), LNR (i.e. the ratio of lymph nodes with tumour metastasis to total lymph nodes analysed), TNM classification, and presence of vascular and perineural invasion. This cohort has been used for pY397 FAK IHC staining. Statistical analyses: Percentage of cells expressing low, medium and high pY397 FAK staining was

quantified using Definiens Tissue Studio® (Imag'IN core Institut Universitaire du Cancer, Toulouse, France) in stromal and tumour areas. Definiens Tissue Studio multi-resolution segmentation algorithm has been applied on PDAC TMA using the following steps. (i) ROI segmentation was identified by image partitionalization into spectrally coherent regions (based on morphology, haematoxylin intensity, area and shape). (ii) Manual classification of subsets of super-pixels (as, e.g., tumour area, stroma area) was performed. (iii) Machine-learning algorithms for automatic morphological segmentation were ran. Software automatically segments each individual cell based on morphology, haematoxylin intensity, area and shape, and identifies each cell as a mathematical object within the different subregions defined previously allowing the nuclear/cytoplasm detection. (iv) IHC staining classification built-in stain deconvolution algorithm enables to separate IHC stain from the background haematoxylin counterstain and to classify it based on intensity into four groups (negative, low, medium or high intensity). FAK activity score (h-score) was calculated based on Definiens results as followed: 1X (%low pY397 FAK cells) + 2X (%medium pY397 FAK cells) + 3X (%high pY397 FAK cells). Patients were separated in half (60/60) based on their pY397 FAK h-score (either in CAFs or in tumour cells depending on the performed analysis), generating pY397 FAK low group (lower half based on the median) and pY397 FAK high group (upper half) of 60 patients each. The outcome variables were DFS and OS. DFS was defined according to the DATECAN consensus (Bonnetain et al, 2014) as the time interval between the day of surgical resection and the date of local relapse, or regional relapse, or occurrence of distant metastases, or appearance of a second PDAC, or death (all causes), whichever occurred first. Survival data were censored at the last follow-up. OS was defined as the time interval between day of surgical resection and death (all causes) or date of last follow-up, at which point data were censored. Survival curves were estimated with the Kaplan–Meier method and compared using log-rank test. We used non-parametric tests (chi-square test) to compare independent groups for categorical data. Multivariate analyses, with backward variable selection, were conducted with the Cox proportional-hazards regression model. Variables up to the 0.05 level in univariate analysis were included in the multivariate model. The level of significance for all tests was defined as $P < 0.05$. Statistical analyses were carried out with PRISM 5 and XLSTAT 2017.

### 47 patient cohort, whole slide

Forty-seven patients were selected from a retrospective cohort of patients with localized (non-metastatic) PDAC who underwent complete surgical resection between December 2011 and January 2014 at Beaujon University Hospital (Clichy, France; Biobank registration number BB-0033-00078) (Neuzillet et al, 2019). All participants gave informed consent before taking part. Exclusion criteria were neoadjuvant chemotherapy and/or radiotherapy, macroscopically incomplete resection (R2), tumour histology other than PDAC, and insufficient tumour material available for research.

### Fresh PDAC patient samples

Tumours from patients with surgery-resected pancreatic cancer were from the Pathology Department of Toulouse and Limoges Hospitals, France. hCAF cultures were obtained from fresh human

PDAC samples after informed consent in accordance with the declaration of Helsinki and stored at the "Cancer Research Center of Toulouse" collection (CRCT, INSERM U1037, No. CPP DC-2016-2654), according to the French law (declaration to the ministry of higher education and research). PDAC patient samples were either paraffin-embedded for IHC staining or processed for CAF isolation (see "Cells" section). This study was approved by the ethic committee of the Institution.

### Cells

#### Human cancer-associated fibroblasts

CAFs were isolated from human pancreatic tumour tissues using the outgrowth method described by Bachem et al (1998). Briefly, human CAFs were isolated using explant techniques from histologically fibrotic areas of PDAC surgically resected. Small tissue blocks were cut (0.5–1 mm³) using razor blade and seeded in 10-cm² uncoated culture wells in the presence of Dulbecco's modified Eagle's medium F12 (DMEM/F12 Lonza D8437) containing 10% FBS (Life Technologies) supplemented with L-Glutamine (L-glu, 2 mmol/l), penicillin and streptomycin (P/S). Tissue blocks were cultured at 37°C in a 5% $CO_2$-air humidified atmosphere. Eighteen hours after seeding, culture medium was changed. CAFs grew out from the tissue blocks 1–3 days later.

#### Mouse pancreatic tumour cells

Mouse pancreatic tumour cells are tumour cells isolated from LSL-K-Ras$^{G12D/+}$;LSL-p53$^{R172H/+}$;Pdx1-Cre (also referred as KPC) (Olive et al, 2009) and generously provided by Dr. D. Saur (University Hospital Klinikum rechts der Isar, Technische Universität München, Germany). Lentivirus-transduced mCherry-expressing tumour cells were puromycin selected and sorting for mCherry. Mouse tumour cells were maintained in DMEM 10% FBS plus P/S and L-glutamine.

#### Mouse embryonic FAK-WT or FAK-KD fibroblast generation

Primary FAK-KD/KD and FAK-WT/WT MEFs were generously provided by Dr. David Schlaepfer (UCSD, San Diego, USA). MEFs were isolated from E8.5 embryo explant culture. Kinase inactive FAK was generated by knock-in point mutation strategy (FAK lysine 454 to arginine) in fak exon 21 (Lim et al, 2010). MEFs were expanded and maintained on dishes pre-coated with 0.1% gelatin (Sigma-Aldrich, #G2500) in DMEM/F12 10% FBS plus P/S and L-glu. After expansion and limited passage, primary MEFs were immortalized via retrovirus-mediated expression of human telomerase reverse transcriptase (hTERT) followed by puromycin selection, or spontaneously immortalized. GFP-MEFs (green-labelled MEFs) were generated by lentiviral infection and GFP sorted.

#### Macrophages derived bone marrow monocytes

Mice were euthanized in accordance with institutional animal ethics guidelines. Femurs and tibias were harvested and bone cavity flushed with serum-free DMEM. Cells were spun and washed and plated in DMEM/F12 10% FBS plus P/S and L-glu. BM cells were treated with murine recombinant M-CSF (20 ng/ml; Immunotools, #12343115) for 10 days in a humidified incubator at 37°C with 5% $CO_2$. Cells were then stimulated with LPS (10 ng/ml; Sigma-Aldrich, L2880-10MG) plus IFN-γ (20 μg/ml;

Peprotech, #315-05-100UG) or with IL-4 (10 μg/ml; Peprotech, #214-14-20UG) plus IL-13 (20 μg/ml; Peprotech, #210-13-20UG) for 24 h in order to polarize M0 macrophages into, respectively, M1 or M2-like macrophages.

## Tumour mouse model: Pancreatic cancer cell and fibroblast syngeneic and orthotopic co-grafting

Mouse pancreatic cancer cells and FAK-WT or FAK-KD fibroblasts were trypsinized, washed and suspended in sterile PBS. A 1:3 mix of pancreatic cancer cells ($20 \times 10^3$) and FAK-WT or FAK-KD fibroblasts ($60 \times 10^3$) contained into 20 μl of PBS were injected in the pancreas of anesthetized (Isoflurane, Isovet from Piramal Health) 8-week-old female c57bl/6j (Charles River, France). Each group was composed of at least five mice. Twenty-one or 38 days after injection, mice were euthanized and primary pancreatic tumour, lung, liver and spleen were removed, analysed and paraffin-embedded before being sliced and stained. Mice were kept under specific pathogen-free conditions. All experiments were in accordance with institutional guidelines and European animal protection law and approved by the responsible government agency (agreement number 201612011806414).

## Colorations and Immunostaining (see Table EV3 for antibody and reagent references)

Paraffin-embedded tissue sections were incubated 30 min at 60°C before being deparaffinized and rehydrated. For H&E staining, slides were then incubated 3 min in haematoxylin, washed 5 min in water, incubated 3 min in eosin and twice soaked in 100% ethanol before being incubated in xylene for 5 min and mounted using Eukitt. For immunostaining, after rehydration, slides were processed for antigen retrieval (10 mM sodium citrate pH6 or Tris-EDTA pH9 and autoclaved at 120°C for 12 min) and quenched for peroxidase activity (10 min with 0.3% hydrogen peroxide at RT) before being washed 5 min in PBS. Periphery of section was marked with hydrophobic marker and sections blocked with blocking buffer during 45 min (RT). Slides were incubated with primary antibodies (see Table EV3) overnight at 4°C. Slides were PBS washed and incubated with ImmPRESS secondary antibody 1 h at RT. After PBS washes, antibody binding was visualized with DAB substrate kit or AEC substrate chromogen and counterstained with haematoxylin (2 min). Slides were washed with water, and sections were mounted using glycergel and imaged using Panasonic 250 scanner (Imag'IN core Institut Universitaire Cancer Toulouse, France). For Sirius Red staining, deparaffinized and hydrated sections were incubated with Pico-Sirius Red Solution for 1 h, rinsed, dehydrated in alcohol and mounted using Eukitt. Sections were analysed using Cell Observer videomicroscope (Zeiss) with polarizer D, 90° rotatable for Axio Observer (Zeiss). According to the manufacturer's instructions (Abcam, ab150681), areas of yellow-orange birefringent fibres corresponding to type I (thick fibres) collagen and green birefringent fibres corresponding to type III (thin fibres) collagen were quantified using ImageJ. Areas of lung metastasis were quantified based on CK19 staining allowing tumour cell identification. Percentage of metastasis area is the ratio of 100* (CK19-positive area divided by total lung area).

To demonstrate the specificity of the FAK phosphorylation site specific antibody, anti-pY397 FAK antibody (clone 31H5L17, Thermo Fisher Scientific) was pre-incubated with 200-fold molar excess of peptide (ab40145) for 12 h rolling at 4°C before being used. In parallel, human PDAC sample slides were pre-incubated with Lambda phosphatase (NEB, P0753S) for 2 h at 37°C (according to the manufacturer's instructions) before being incubated with FAK Y397 antibody (clone 31H5L17, Thermo Fisher Scientific).

## Immunofluorescence staining (see Table EV3 for antibody and reagent references)

Human primary CAFs were plated in petri dish with coverslips previously coated with 0.1% gelatin (Sigma-Aldrich). After cell adhesion, cells were starved with DMEM/F12 0.5% FBS overnight. hCAFs were then treated during 48 h with FAK inhibitor (PF-562271, Selleckchem, #S2890) at 1 μM. Forty-eight hours later, coverslips were processed for IF staining. Cells were first fixed (10 min, 10% formalin), permeabilized (10 min, 0.05% Triton X-100) and incubated with 1% BSA for 1 h. Primary antibodies for periostin, osteopontin, lysyl oxidase-like 2 (LOXL2), pY397 FAK, collagen I, collagen III, collagen IV, FAK and secreted protein acidic and cysteine rich (SPARC) were added in 1% BSA overnight at 4°C. Cells were PBS washed and DAPI plus secondary antibodies Goat anti-Mouse Alexa Fluor-488 or Goat anti-Rabbit Alexa Fluor-647 were added for 1 h at RT. After PBS washes, slides were mounted in Fluorescent mounting Medium and images acquired using Cell Observer videomicroscope (Zeiss), Zen 2012 blue edition. At least 20 cells per patient and per condition were imaged. Quantification was performed using ImageJ. Files were cropped and contrast adjusted using Photoshop (Adobe).

## Western blot (see Table EV3 for antibody and reagent references)

Cells were lysed in lysis buffer (50 mM Hepes pH 7.4, 150 mM NaCl, 10% glycerol, 1% Triton X-100, 0.1% SDS, 1% sodium deoxycholate, 5 mM sodium orthovanadate, 2 mM NaF, 1 mM DTT and a cocktail of protease inhibitors, Roche). Protein extract concentration was measured using Protein Assay Reagent (Bio-Rad), and equal amounts of proteins were resolved by SDS–PAGE and transferred onto nitrocellulose membranes. Membranes were blocked in 5% BSA in Tris-buffered saline with 0.1% Tween 20 (TBST) for 1 h followed by incubation with primary antibodies overnight at 4°C. Membranes were then washed with TBST three times and incubated with horseradish peroxidase-coupled secondary antibody for 1 h at room temperature, washed again and treated with enhanced chemiluminescence prior to detection on ChemiDoc Imaging System (Bio-Rad).

To demonstrate the specificity of the FAK phosphorylation site specific antibody, anti pY397 FAK antibody (clone 31H5L17, Thermo Fisher Scientific) was pre-incubated with the synthetic blocking peptide (ab40145) for 12 h rolling at 4°C before being used as described previously. In parallel, fibroblast lysate (50 μg) was pre-incubated either with Lambda phosphatase (NEB, P0753S) overnight at 37°C (according to the manufacturer's instructions) or with 5 mM sodium orthovanadate before being processed for Western blotting using FAK Y397 antibody (clone 31H5L17, Thermo Fisher Scientific).

## Scratch wound healing assay

$25 \times 10^3$ red-labelled pancreatic tumour cells (mCherry-transfected cells) and $25 \times 10^3$ FAK-WT or FAK-KD green-labelled fibroblasts (GFP-transfected cells) were mixed and plated in 0.1% gelatin pre-coated Lab-Tek glass chamber (Lab-Tek, #154941) in DMEM/F12 complete medium overnight. Cells were then starved with 0.5% FBS plus 0.5 μg/ml Mitomycin C for 1 h. Confluent monolayer was then scratched using sterile 10 μl pipette tip. Cells were PBS washed before DMEM/F12 plus 0.5% FBS and 0.5 μg/ml mitomycin C added. Images were acquired every 3 min for 72 h at 37°C and 5% $CO_2$ on Cell Observer videomicroscope motorized with AxioObserver Z1 (ZEISS); Objective 10× EC Plan-Neofluar; Source Colibri 2LED (ZEISS), using software Zen Blue 2012. Scratch wound assay movie was made from time zero to 72 h. Individual cell tracking was performed using MTrackJ plugin of ImageJ. Distance of migration (length of the path travelled), directionality (calculated by comparing the Euclidian distance to the accumulated distance represents a measurement of the directness of cell trajectories) and migration velocity (rapidity of cell motion, calculated on moving cells) were analysed using Chemotaxis and Migration Tool software (free software tool for data analysis from time stack chemotaxis experiments, based on the National Institute of Health's (NIH) ImageJ image processing system).

## ECM analyses by NHS staining

To promote ECM generation, 0.1% gelatin pre-coated coverslips were treated with 1% glutaraldehyde in PBS for 30 min and washed before being treated with 1 M ethanolamine (30 min, RT). Fibroblasts were then plated (DMEM/F12 0.5% FBS) and activated in presence of 8 ng/ml TGF-β. Confluent monolayer was scratched using sterile 200 μl pipette tip, and cells were washed (PBS) and new medium added (DMEM/F12 0.5% FBS + TGF-β). The medium was changed every 2 days (with fresh TGF-β) until scratch closing (8 days). At day 8, samples were decellularized using alkaline detergent extraction buffer (0.5% Triton X-100, 20 mM $NH_4OH$ in PBS). After 2 min, coverslips were washed with PBS, and DNase treated (10 μg/ml) for 30 min at 37°C. To visualize the ECM, the coverslips were labelled with 2 μg/ml NHS-ester Alexa Fluor 488 dye (A20000; Thermo Fisher Scientific) diluted in 50 mM sodium bicarbonate buffer pH 9 for 15 min in the dark. Coverslips were washed, treated with 200 mM Tris buffer pH 7.5 for 10 min to deactivate the NHS esters and blocked in 1% BSA before being mounted in Fluorescent Mounting Medium. Images were acquired using Cell Observer videomicroscope (Zeiss), Zen 2012 blue edition. Files were cropped and contrast adjusted using Photoshop (Adobe).

## 3D invasion assay

3D spheroids of FAK-WT or FAK-KD fibroblasts plus pancreatic tumour cells were generated using the hanging drop technique. A 20 μl drop containing 5,000 FAK-WT or FAK-KD green-labelled fibroblasts (GFP-transfected cells) plus 5,000 red-labelled pancreatic tumour cells (mCherry-transfected cells) was hanged for 4 days at 37°C on a 24-well plate lid. New formed spheroids were transferred into well (eight-well bottomless μ-Slide Ibidi, #80828) pre-coated with 50 mg/ml collagen I matrix (type I collagen, Corning, #354249)

and embedded into 2 mg/ml collagen I matrix for 72 h. Z-stack of embedded spheroids was made using Zeiss LSM 780 confocal microscope every 24 h at 10× magnification (every 3 μm). Analyses and 3D reconstitution were made with Zen 2012 Blue edition. Quantification was performed using ImageJ. Distance of cell invasion represents the distance between the spheroid edge and the cell.

## Collagen matrix contraction

Collagen lattices were prepared using type I collagen (Corning, #354249) according to the manufacturer's instructions. FAK-WT or FAK-KD fibroblasts and hCAFs were trypsinized, counted, pre-treated or not with FAK inhibitor (1 μM) for 1 h (treated with TGF-β at 2 ng/ml, if needed) before cell-matrix-embedment. Cell-collagen-embedment: fibroblasts alone (hCAFs or FAK-WT or -KD fibroblasts, $2 \times 10^5$ cell/ml) or fibroblasts plus tumour cells ($10^5$ cell/ml of fibroblasts plus $10^5$ cell/ml of tumour cells) were suspended in collagen (1 mg/ml of collagen final concentration) and aliquoted into 24-well plates (500 μl/well). Collagen lattices were polymerized for 1 h at RT. To initiate collagen gel contraction, polymerized gels were gently released from the underlying culture dish and DMEM/F12 0.5% FBS medium was immediately added in presence or not of FAK inhibitor (PF-562271 at 1 μM) and/or TGF-β (2 ng/ml). Medium was changed every 2 days. The degree of collagen gel contraction was analysed during 7 days. Gel diameter was measured in mm and recorded as the average values of the major and minor axes. Gel contraction = 100*(well diameter − gel diameter)/well diameter.

## Flow cytometry (see Table EV2 for antibody and reagent references)

### CAF marker analyses

Thirty-eight days after grafting, c57bl/6j mice were euthanized and primary pancreatic tumour resected. Tumours were dissociated using Tumor Dissociation Kit (130-096-730, MACS, Miltenyi Biotec) according to the manufacturer's instructions. Cells were then fixed and permeabilized (Fix/Perm solution) for 30 min at RT. Cells were washed before being incubated with pY397 FAK, or FAPα antibodies (30 min, RT) diluted into permeabilization buffer (PB). After PB washes, cells were incubated with secondary antibodies: anti-rabbit Alexa Fluor-647, anti-rabbit Alexa Fluor-546 or anti-mouse Alexa Fluor-647 for 15 min at RT. After PB washes, cells were incubated with PDGFR-α-PE or α-SMA-Alexa Fluor 647 antibodies for 30 min at RT, washed and staining analysed using Fortessa X20 flow cytometer (BD Biosciences) and DIVA version 6.1.2 software (BD Biosciences).

### Immune cell marker analyses

Thirty-eight days after grafting, mice were euthanized and single-cell suspensions were prepared as for "CAF marker analyses". Digestion mixtures were filtered through 40-μm nylon strainers (Fisher Scientific). Cells were washed twice in FACS buffer (PBS, BSA 1%, FBS 10%), counted and incubated with FcReceptor-blocking for 30 min at RT. For extracellular staining, cells were then incubated for 30 min at RT with indicated antibodies (see Table EV3). Cells were washed and staining analysed. For intracellular staining, cells were fixed, permeabilized (Fix/Perm solution) for 30 min at RT, washed with PB, and incubated with FOXP3-PE and Nos2-PECy5 for 30 min at RT, washed and staining analysed.

All stained cells were analysed using Fortessa X20 flow cytometer (BD Biosciences) and FlowJo V10 software (BD Biosciences).

### Ex vivo macrophage polarization analyses

Bone marrow-derived monocytes were collected from mice bones (tibia). After recovering the marrow by injecting DMEM/F12 10% FBS + P/S + glu in bone holes with a 5 ml syringe and a 12 mm needle, cells were dissociated with a 1,000 μl pipette and filtered using a 40-μm cell strainer. $4.10^6$ cells were plated in petri dish in DMEM/F12 + P/S + glu with 20 ng/ml M-CSF to differentiate monocytes into macrophages for 10 days. After 3 days, the supernatant (containing dead and unadhered cells) was centrifuged at 800 rpm 10 min to remove dead cells and the monocytes were re-plated in fresh medium with M-CSF. At day 10, M0 macrophages were treated for 24 h either with 20 ng/ml M-CSF for M0 macrophages as control, or 10 ng/ml IL-4 + 20 ng/ml IL-13 for M1 macrophages as control, or 20 ng/ml IFNγ + 10 ng/ml LPS for M2 macrophages as control, or CM from FAK-WT or FAK-KD activated fibroblasts. Macrophages were washed and detached using a Versene solution (#15040066, Thermo Fisher Scientific) and CD163, CD206, F4/80, CD11b, CD80, CD86, dectin, CHMII cell markers analysed as previously described.

### CAF conditioned medium stimulation

Human primary pancreatic CAFs were cultured in DMEM/F12 0.5% FBS overnight before being treated with new DMEM/F12 0.5% FBS ± FAK inhibitor (PF-562271, Selleckchem, #S2890) at 1 μM for 48 or 72 h. Mouse tumour cells were cultivated a week with 45% DMEM/F12, 45% DMEM and 10% FBS, then a week with DMEM/F12 10% FBS, and finally, cultured in serum-free DMEM/F12 for 48 h. CM from either CAFs or tumour cells were then harvested and filtered through 0.22-μm filter (Millipore; SLGS033SS).

### Transwell migration assay

Transwell chambers (Corning, NY, #3421) containing 6.5 mm Transwell® with 5.0 μm Pore Polyester Membrane Insert were used for cell migration assay. A total of $1 \times 10^5$ macrophages were seeded on the upper compartment, and CM from CAFs pre-treated or not with FAK inhibitor on the lower compartment of the transwell chamber. Macrophages were allowed to migrate through the filter for 24 and 48 h. Cells from the lower compartment were then stained with crystal violet and quantification performed using ImageJ based on images acquired using Cell Observer videomicroscope (Zeiss), Zen 2012 blue edition.

### RT–qPCR

RT–qPCR was performed on hCAF mRNAs isolated using TRIzol-LS (Thermo Fisher Scientific), and concentration measured using a NanoDrop (ND-1000 from Thermo Fisher Scientific). RNAs were first reverse-transcribed with RevertAid H Minus Reverse Transcriptase (Invitrogen) using random hexamer primers (Invitrogen) according to the manufacturer's instruction, qPCR was carried out on a StepOne Plus (Applied Biosystems, Life Technologies) using SsoFast EvaGreen Supermix (Bio-Rad) and primer concentration at 0.5 μM. Primers specific to human mRNAs were purchased from

Integrated DNA Technologies (IDT) and are listed on the following table:

| | Primer forward | Primer reverse |
|---|---|---|
| hCCL2-1 | 5′-CAGCCAGATGCAATCAATGCC-3′ | 5′-TGGAATCCTGAACCCAC TTCT-3′ |
| hCCL2-2 | 5′-CCCAAAGAAGCTGTGATCTTCA-3′ | 5′-TCTGGGGAAAGCTAGG GGAA-3′ |
| hCCL3 | 5′-CTCTGCAACCAGTTCTCTGC-3′ | 5′-TCGCTTGGTTAGGAAGA TGACA-3′ |
| hCCL4 | 5′-CTCCCAGCCAGCTGTGGTATTC-3′ | 5′-CCAGGATTCACTGGGAT CAGCA-3′ |
| hCCL5 | 5′-CGTGCCCACATCAAGGAGTA-3′ | 5′-TCGGGTGACAAAGACG ACTG-3′ |
| hM-CSF | 5′-CGGGACCCAGCTGCC-3′ | 5′-TGACTGTCAATCAGC CGC-3′ |

### ELISA

Levels of MCP-1 were quantified by using the Human MCP-1 (CCL2) Pre-Coated ELISA Kit (BGK13500) according to the manufacturer's instructions (Biogems, PeproTech Brand). O.D. absorbance was measured at 450 nm using Mithras LB940 (Berthold Technologies).

### Matrisome

#### Matrix production and extraction

Matrix production and extraction was performed as in (Kaukonen *et al*, 2017). Briefly, CAFs were seeded in a 12-well plate, and once confluent, medium was replaced by fibroblast growth medium supplemented with ascorbic acid (50 μg/ml) and treated or not with FAK inhibitor (1 μM) every other day for 7 days. Cells were then washed with PBS before being treated with pre-warmed sterile-extraction solution (20 mM $NH_4OH$, 0.5% Triton X-100 in PBS) for 1 min. Extraction buffer was removed immediately by gently pipetting (avoiding to disrupt the matrix). Cell-derived matrices were washed twice with PBS before being treated with DNase I (10 μg/ml) for 30 min at 37°C and then washed again twice with PBS.

#### Peptide preparation

Peptide preparation was performed as in (Pearce *et al*, 2018). ECM was then scrapped into Urea Buffer (250 μl of an 8 M Urea in 20 mM HEPES [pH8]) solution containing $Na_3VO_4$ (100 mM), NaF (0.5 M), β-Glycerol Phosphate (1 M), $Na_2H_2P_2O_7$ (0.25 M) and transferred into Eppendorf. Samples were vortexed for 30 s and left on ice prior to sonication at 50% intensity, three times for 15 s on ice. Centrifuge tissue lysate suspension at 20,000 *g* for 10 min at 5°C, and recover supernatant to a protein LoBind tube. BCA assay for total protein was then performed and 80 μg of protein continued to the next step. Volume of supernatant was adjusted to 200 μl. Prior to trypsin digestion, disulphide bridges were reduced by adding 500 mM dithiothreitol (DTT, in 10 μl) to sample(s), vortexed and then incubated at RT for 1 h with agitation in the dark. Free cysteines were then alkylated by adding 20 μl of a 415 mM iodoacetamide solution to sample and then vortexed and incubated at RT for 1 h with agitation in the dark. The sample was then diluted one

in four with 20 mM HEPES. Removal of *N*-glycosylation was then achieved by addition of 1,500 U PNGaseF, vortexed and incubated at 37°C for 2 h. Two microliter of a 0.8 µg/µl LysC sample was then added, gently mixed and then incubated at 37°C for 2 h. Protein digestion was then achieved with the use of immobilized Trypsin beads (40 µl of beads per 250 µg of protein) and incubation at 37°C for 16 h with shaking performed. Peptides were then de-salted using Glygentop tips C-18. Briefly, samples were acidified with trifluoroacetic acid (1% v/v). Samples were then centrifuged at 2,000 *g*, 5 min at 5°C and transfer supernatant to a 1.5 ml Eppendorf. Glygentops tip was washed with 200 µl of 100% ACN (LC-MS grade), centrifugation (1,500 *g* for 3 min) performed, flow-through discarded and washed with 99% $H_2O$ (+1% ACN, 0.1% TFA) twice. Protein digest sample were loaded, centrifuged and flow-through discarded. Cartridge was washed with 200 µl of 99% $H_2O$ (+1% ACN, 0.1% TFA). Centrifugation and flow-through discarded were performed. The tip was transferred into a new LoBind Eppendorf and 100 µl of 70/30 ACN/$H_2O$ +0.1% FA added. Centrifugation was performed and flow-through kept. Samples were dried and stored at −20°C. Re-constitute samples in re-con buffer prior to MS analysis. Mass spectrometry analysis and bioinformatics: Dried samples were dissolved in 0.1% TFA (0.5 µg/µl) and analysed by nanoflow ultimate 3000 RSL nano instrument that was coupled on-line to a Q Exactive plus mass spectrometer (Thermo Fisher Scientific). Peptides were separated using analytical column (EASY-Spray; Cat. #ES803) and using solvent A (0.1% FA in LC–MS grade water) and solvent B (0.1% FA in LC–MS grade ACN) as mobile phases. The UPLC settings consisted of a sample loading flow rate of 12 µl/min for 3 min followed by a gradient elution with starting with 3% of solvent B and ramping up to 28% over 120 min followed by a 7 min wash at 80% B and a 10 min equilibration step at 3% B. The flow rate for the sample run was 250 nl/min with an operating back pressure of about 550 bar. Full scan survey spectra (*m/z* 375–1,500) were acquired with a resolution a 70,000 FWHM resolution. A data-dependent analysis (DDA) was employed in which the 15 most abundant multiply charged ions present in the survey spectrum were automatically mass-selected for HCD (higher energy collisional dissociation) and MS/MS scanning (200–2,000 *m/z*) with a resolution of 17,500 FWHM. A 30 s dynamic exclusion was enabled with the exclusion list restricted to mass window of 10 ppm. Overall duty cycle generated chromatographic peaks of approximately 30 s at the base, which allowed the construction of extracted ion chromatograms (XICs) with at least 10 data points.

### Data processing protocol

MASCOT search was used to generate a list of proteins. Peptide identification was by searchers against the Swiss-Prot database restricted to human entries using the Mascot search engine (v 2.6.0, Matrix Science, London, UK). The parameters included trypsin as digestion enzyme with up to two missed cleavages permitted, carbamidomethyl (C) as a fixed modification and Pyro-glu (N-term), Oxidation (M) and Phospho (STY) as variable modifications. Datasets were searched with a mass tolerance of ± 5 ppm and a fragment mass tolerance of ± 0.8 Da. A MASCOT score cut-off of 50 was used to filter false-positive detection to a false discovery rate below 1%. PESCAL was used to obtain peak areas in extracted ion chromatograms of each identified peptide (Cutillas & Vanhaesebroeck, 2007) and protein abundance determined by the ratio of the sum of peptides areas of a given protein to the sum of all peptide areas. This approach for global protein quantification absolute quantification, described in (Cutillas & Vanhaesebroeck, 2007), is similar to intensity-based protein quantification (iBAQ) (Schwanhausser *et al*, 2011) and total protein abundance (TPA) (Wisniewski *et al*, 2012). The mass spectrometry proteomic data have been deposited to the ProteomeXchange Consortium via the PRIDE partner repository with the dataset identifier PXD018899.

### β1-integrin activation (see Table EV3 for antibody and reagent references)

Human primary CAFs were plated in coverslips coated with 0.1% gelatin and treated with glutaraldehyde/ethanolamine to favor matrix generation as for NHS staining (see above). After adhesion, cells were starved with DMEM/F12 0.5% FBS overnight before being treated with FAK inhibitor (PF-562271, Selleckchem, #S2890) at 1 µM during 7 days (medium was changed every 2 days with fresh inhibitor). Coverslips were decellularized with an extraction buffer as for NHS staining and tumour cells were plated on the matrices at 20% confluence for 16 h. The negative and positive controls were done by adding 5 mM EDTA and 1 mM $MnCl_2$, respectively, for 20 min at 37°C before being processed for imaging. Cells were fixed (15 min, 3.7% paraformaldehyde), permeabilized (10 min, 0.1% Triton X-100) and incubated with 2% BSA for 1 h. Antibodies against β1-integrin (#102207, BioLegend) or activated β1-integrin (CD29 9EG, 553715, BD Biosciences) were added in 2% BSA 2 h at RT. Cells were PBS washed and incubated with DAPI (1/1,000) plus secondary antibodies Goat anti-Rat Alexa Fluor-555 and Phalloidin-Alexa Fluor 488 for 45 min at RT. After PBS washes, slides were mounted in Fluorescent Mounting Medium and images acquired using Confocal microscope (Zeiss), Zen 2012 blue edition. Quantification was performed using ImageJ. Files were cropped and contrast adjusted using Photoshop (Adobe). Quantification was performed using ImageJ.

### Statistical analyses

Classical statistical analysis was run using the GraphPad Prism software (GraphPad). Data were tested to confirm the Gaussian distribution (Shapiro–Wilk test) and the similarities of variances (*F*-test or chi-square test). Depending on experiments, two-group data were analysed using unpaired or paired, one or two-tailed Student *t*-test (Gaussian distribution and equal variances) or Mann–Whitney test (non-Gaussian distribution). Multi-group data were analysed using one-way ANOVA followed with the Tukey's multiple comparison post-test (Gaussian distribution). All values are mean ± SEM. Differences were considered statistically significant when $P < 0.05$ (*$P < 0.05$, **$P < 0.01$, ***$P < 0.001$). Exact *P*-values are indicated for main and EV figures in the Appendix Table S1.

Survival curve statistical analyses were estimated with the Kaplan–Meier method and compared using log-rank test. We used non-parametric tests (chi-square test) to compare independent groups for categorical data. Multivariate analyses, with backward variable selection, were conducted with the Cox proportional-hazards regression model. Variables up to the 0.05 level in univariate analyses were included in the multivariate model. The level of significance for all tests was defined as $P < 0.05$. Statistical analyses were carried out with PRISM 5 and XLSTAT 2017.

## The paper explained

### Problem

Understanding how cancer-associated fibroblasts (CAFs), the most abundant pancreatic ductal adenocarcinoma (PDAC) stromal cells, promote tumour growth, disease progression and resistance to therapies is of major interest, given the poor prognosis and survival of pancreatic cancer patients. This study identifies a druggable key regulator of CAF-induced tumour progression and a prognostic factor for the disease: the protein tyrosine kinase focal adhesion kinase (FAK).

### Results

Using two independent cohorts including over 160 PDAC samples, we identify FAK activity (pY397 FAK score) to be significantly increased in CAFs from PDAC compared to fibroblasts from healthy pancreas, and we report that high fibroblastic FAK activity in PDAC tumours is an independent prognostic factor for disease-free survival (DFS) and overall survival (OS) in multivariate analysis. We demonstrate that fibroblastic FAK activity is a potent driver of tumour cell invasion, being involved in CAF-dependent pancreatic tumour migration and invasion, in ECM remodelling and in immunosuppression. Genetic FAK inactivation within CAFs modifies primary tumour stroma (ECM and immune cell population) and drastically reduces spontaneous metastasis in an orthotopic and syngeneic mouse model.

### Impact

Our study identifies FAK activity within fibroblasts as a key regulator of PDAC progression and as an independent prognostic marker for patients. It highlights the potential benefit of evaluating FAK activity within CAFs to identify patients with high risk of early relapse and patients that may benefit from a personalized protocol using FAK inhibitor treatment, currently used in clinical trials.

## Data availability section

The datasets and computer code produced in this study are available in the following database: Matrisome data: PRIDE PXD018899 (http://www.ebi.ac.uk/pride/archive/projects/PXD018899)

**Expanded View** for this article is available online.

## Acknowledgements

We thank Dr Frédéric Lagarrigue (IPBS, Toulouse) for his help with beta-1 integrin assay, and Marcin Domagala and Sophie Gazzola (CRCT, Toulouse) for their advice on macrophage experiments. We thank Laetitia Ligat (CRCT), Sophie Allart (CPTP) and Astrid Canivet-Laffitte (CPTP) for their help and expertise in microscopy; Manon Farce (CRCT) for her help and flow cytometry expertise; and Loic Van Den Berghe and Christelle Segura for their help in lentivirus production and expertise in vectorology. The authors also acknowledge Dr Francois-Xavier Frenois from the Imag'IN platform (IUC, Toulouse, France) for his help and expertise on Definiens Tissue Studio® and Dr Pedro Cutillas from the Centre for Haemato-Oncology (Barts Cancer Institute, Queen Mary University of London, UK). We would like to thank Servier Medical Art (https://smart.servier.com) for graphics used in the figures. This work was supported by LNCC (Ligue Nationale Contre le Cancer RAB20007BBA, RAB20008BBA and RAB17029BBA), the Canceropole Grand Sud-Ouest (RMA04002BPA), the Labex TOUCAN, the Toulouse University IDEX (G16001BB), the PHUC CAPTOR, la Fondation Bristol-Myers Squibb and the French National Institute of Cancer (INCa, PLBIO2015-115, PAIR 2018-080). I.B is a recipient of LNCC fellowship (IP/SC16060). E.D. and J.R. salaries are funded, respectively, by Fondation pour la Recherche Médicale (FRM ING20150532688) and the French National Institute of Cancer (PLBIO2015-115); R.S. was a recipient of a fellowship from FRM (#40493). C.J. was recipient of a fellowship from the LNCC. O.M.T.P is a recipient of a Cancer Research UK and Credit Suisse fellowship (A27947).

## Author contributions

SZ, ED, IB, RS, SCS, JR, AB, MS, AA, AP, MM and CJ performed the experiments. SZ, ED, IB and CJ analysed and interpreted the data. JC and CN provided data collection. RT, AP and MM were involved in fresh sample collection and CAF isolation. VR and OMTP were involved in matrisome experiment. SZ, CB and CJ designed the study and wrote the manuscript. DS, YM and SP contributed to the result interpretation. All authors critically reviewed and approved the final manuscript.

## Conflict of interest

The authors declare that they have no conflict of interest.

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
