## [Review Process File · EMBO Molecular Medicine]

FAK activity in cancer-associated fibroblasts is a prognostic marker and a druggable key metastatic player in pancreatic cancer

Sonia Zaghdoudi, Emilie Decaup, Ismahane Belhabib, Rémi Samain, Cassant-Sourdy Stéphanie, Julia Rochotte, Alexia Brunel, David Schlaepfer, Jérôme Cros, Cindy Neuzillet, Manon Strehaiano, Amandine Alard, Richard Tomasini, Vinothini Rajeeve, Aurélie Perraud, Muriel Mathonnet, Oliver Pearce, Yvan Martineau, Stéphane Pyronnet, Corinne Bousquet, and Christine Jean
DOI: [10.15252/emmm.202012010](https://doi.org/10.15252/emmm.202012010)

Corresponding authors: Christine Jean (christine.jean@inserm.fr), Corinne Bousquet (corinne.bousquet@inserm.fr)

Review Timeline:

Submission Date:	14th Jan 20
Editorial Decision:	11th Feb 20
Revision Received:	10th Jul 20
Editorial Decision:	14th Aug 20
Revision Received:	4th Sep 20
Accepted:	8th Sep 20

Editor: Lise Roth

Transaction Report:

11th Feb 2020

Dear Dr. Jean,

Thank you for the submission of your manuscript to EMBO Molecular Medicine. We have now received feedback from the three reviewers who agreed to evaluate your manuscript. As you will see from the reports below, the referees acknowledge the interest of the study and are overall supporting publication of your work pending appropriate revisions.

Addressing the reviewers' concerns in full will be necessary for further considering the manuscript in our journal, and acceptance of the manuscript will entail a second round of review. EMBO Molecular Medicine encourages a single round of revision only and therefore, acceptance or rejection of the manuscript will depend on the completeness of your responses included in the next, final version of the manuscript. For this reason, and to save you from any frustrations in the end, I would strongly advise against returning an incomplete revision.

When submitting your revised manuscript, please carefully review the instructions that follow below. Failure to include requested items will delay the evaluation of your revision:

2) Individual production quality figure files as .eps, .tif, .jpg (one file per figure).

3) A .docx formatted letter INCLUDING the reviewers' reports and your detailed point-by-point responses to their comments. As part of the EMBO Press transparent editorial process, the point-by-point response is part of the Review Process File (RPF), which will be published alongside your paper.

4) A complete author checklist, which you can download from our author guidelines (<https://www.embopress.org/page/journal/17574684/authorguide#submissionofrevisions>). Please insert information in the checklist that is also reflected in the manuscript. The completed author checklist will also be part of the RPF.

6) Before submitting your revision, primary datasets produced in this study need to be deposited in an appropriate public database (see <https://www.embopress.org/page/journal/17574684/authorguide#dataavailability>). Please remember to provide a reviewer password if the datasets are not yet public. The accession numbers and database should be listed in a formal "Data Availability" section (placed after Materials & Method). Please note that the Data Availability Section is restricted to new primary data that are part of this study.

7) We would also encourage you to include the source data for figure panels that show essential data. Numerical data should be provided as individual .xls or .csv files (including a tab describing the data). For blots or microscopy, uncropped images should be submitted (using a zip archive if multiple images need to be supplied for one panel). Additional information on source data and instruction on how to label the files are available at .

8) Our journal encourages inclusion of *data citations in the reference list* to directly cite datasets that were re-used and obtained from public databases. Data citations in the article text are distinct from normal bibliographical citations and should directly link to the database records from which the data can be accessed. In the main text, data citations are formatted as follows: "Data ref: Smith et al, 2001" or "Data ref: NCBI Sequence Read Archive PRJNA342805, 2017". In the Reference list, data citations must be labeled with "[DATASET]". A data reference must provide the database name, accession number/identifiers and a resolvable link to the landing page from which the data can be accessed at the end of the reference. Further instructions are available at .

9) We replaced Supplementary Information with Expanded View (EV) Figures and Tables that are collapsible/expandable online. A maximum of 5 EV Figures can be typeset. EV Figures should be cited as 'Figure EV1, Figure EV2' etc... in the text and their respective legends should be included in the main text after the legends of regular figures.

- Additional Tables/Datasets should be labeled and referred to as Table EV1, Dataset EV1, etc. Legends have to be provided in a separate tab in case of .xls files. Alternatively, the legend can be supplied as a separate text file (README) and zipped together with the Table/Dataset file. See detailed instructions here:

10) Every published paper now includes a 'Synopsis' to further enhance discoverability. Synopses are displayed on the journal webpage and are freely accessible to all readers. They include a short stand first (maximum of 300 characters, including space) as well as 2-5 one-sentences bullet points that summarizes the paper. Please write the bullet points to summarize the key NEW findings. They should be designed to be complementary to the abstract - i.e. not repeat the same text. We encourage inclusion of key acronyms and quantitative information (maximum of 30 words / bullet point). Please use the passive voice. Please attach these in a separate file or send them by email, we will incorporate them accordingly.

Please also suggest a striking image or visual abstract to illustrate your article. If you do please provide a jpeg file 550 px-wide x 400-px high.

10) For more information: There is space at the end of each article to list relevant web links for further consultation by our readers. Could you identify some relevant ones and provide such information as well? Some examples are patient associations, relevant databases, OMIM/proteins/genes links, author's websites, etc...

11) As part of the EMBO Publications transparent editorial process initiative (see our Editorial at <http://embomolmed.embopress.org/content/2/9/329>), EMBO Molecular Medicine will publish online a Review Process File (RPF) to accompany accepted manuscripts.

In the event of acceptance, this file will be published in conjunction with your paper and will include the anonymous referee reports, your point-by-point response and all pertinent correspondence relating to the manuscript. Let us know whether you agree with the publication of the RPF and as here, if you want to remove or not any figures from it prior to publication.

I look forward to receiving your revised manuscript.

Yours sincerely,

Lise Roth

Lise Roth, PhD
Editor
EMBO Molecular Medicine

To submit your manuscript, please follow this link:

Link Not Available

Photos 400-800 DPI

Figures are not edited by the production team. All lettering should be the same size and style; figure panels should be indicated by capital letters (A, B, C etc). Gridlines are not allowed except for log

plots. Figures should be numbered in the order of their appearance in the text with Arabic numerals. Each Figure must have a separate legend and a caption is needed for each panel.

*Additional important information regarding figures and illustrations can be found at <http://bit.ly/EMBOPressFigurePreparationGuideline>

***** Reviewer's comments *****

Referee #1 (Remarks for Author):

This manuscript shows some exiting data demonstrating a tumour promoting function of high FAK activity in cancer associated fibroblasts (CAFs). The manuscript is structured into two parts. In the first part the authors explore whether the level of FAK activation in cancer associated fibroblasts (CAFs) could be used as a prognostic marker for resected PDAC patients. Therefore, the analysis of two different tissue arrays (TMA) is performed (Figs 1-2). To assess FAK activation, both sets of TMAs were stained using an antibody specific for FAK phosphorylation at tyrosine 397 (pY397), a marker for FAK activation, and fibroblasts were identified based on their morphology using manual scoring or automated image analysis. The authors observe that CAFs show increased FAK activation compared to fibroblasts in tumour free pancreas and that in resected PDAC patients' high levels of FAK activation in CAFs correlates with reduced disease-free survival and reduced overall survival. The analysis is overall well performed and comprehensive.

In the second part, the authors aim to better understand how FAK activation in CAFs promotes PDAC progression by using a variety of in vivo and in vitro experiments. First, the authors use a syngeneic mouse model mouse, whereby KPC derived cancer cells are orthotopically co-injected into the pancreas together with immortalised mouse embryonic fibroblast (MEF) cell lines initially generated from WT and FAK kinase dead (FAK-KD) mutant mice (Fig. 3). This model reveals that tumours containing FAK KD fibroblasts show a reduction in extracellular matrix deposition and a change in tumour associated macrophage polarization towards a pro-inflammatory M1-like phenotype compared to animals co-injected with FAK WT fibroblasts. Most strikingly, co-implantation of FAK KD fibroblasts significantly reduces lung metastasis compared to animals co-implanted with FAK WT fibroblasts. The authors further describe that FAK KD fibroblasts reduce the migration and invasion of tumour cells in vitro (Fig 4) and that pharmacological inhibition of FAK reduces ECM production and deposition of fibroblasts in vitro (Fig 5). In Fig 6, the authors explore the effect of fibroblast conditioned media (CM) collected from fibroblast treated or not with a FAK inhibitor on macrophage activation and migration. They observe that macrophages cultured in CM collected from FAK inhibitor treated fibroblasts show less expression of the M2 marker CD206 and that treatment of fibroblasts with a FAK inhibitor is less potent to attract M2-like macrophages compared ton CM of untreated fibroblasts.

Strength: t

These are all interesting observations describing that high FAK activity in fibroblasts i. correlates with poor prognosis in PDAC patients, ii. promotes ECM production/deposition by fibroblasts, iii. increases cancer cell invasion/metastasis and iv. attracts M2-like macrophages to the site of tumour formation. These findings are novel since the role of FAK in PDAC progression has mainly been studied with a focus on cancer cells and not on the surrounding stroma compartment. The manuscript is well structured and the data is well presented.

Main concerns:

Fig 2B-E: Please explain in more detail how the survival curves displayed in Fig2B-E were generated. The method section describes that patients were scored based on neg/low/med/high pY397 FAK levels as shown in Fig2A. However, the graphs in panel B-E show only two groups (low/high), with n=60 patients each. Were there no patients with medium pY397 FAK levels? Or, do the graphs as displayed compare the upper half versus the lower half of pY397 FAK levels based on the median? Please explain.

Fig 3: Characterization of in vivo model: FAK KD reduces collagen production. Does the altered stroma also affect the architecture of the tumour vascularization which could also impact metastatic spreading? Include stain for vessel density/vessel maturation and hypoxia. Along these lines, since FAK deficient CAFs reduce tumour cell invasion as reported in Fig 4J, please also check for changes in cancer cell invasion at the invasive front of these tumours.

Fig 4: The described reduced directional migration of tumour cells in the presence of FAK KD fibroblasts compared to tumour cells co-culture with FAK WT fibroblasts is very interesting, but the underlying molecular mechanisms remain largely elusive.

As mentioned by the authors, fibroblasts provide invasive tracks made of ECM deposition for tumour cells. Thus, the authors could explore in more detail the molecular composition of these tracks and how FAK activity changes their deposition and ultimately the migration capacity of tumour cells (different integrins involved).

Fig 5: The analysis of ECM production is largely based on IF staining's. The authors should use an additional method to confirm the observed changes in ECM production. The manuscript would benefit of a more comprehensive analysis beside the selected proteins. For example, the authors could apply a proteomic based secretome or ECM deposition analysis to thoroughly assess the function of FAK in ECM protein secretion and or deposition.

Fig 6: Changes in macrophage polarization is assessed based on the two markers CD206 and Nos2. The authors should assess additional M1/M2-like markers to better characterise changes in their activation. Also, as written it appears that murine macrophages were used to assess the effect of human CAF conditioned media on macrophage activation. Some of the secreted factors might not conserve their biological activity across the different species. Also, a more detailed analysis of the fibroblast secretome (+/- FAK inhibitor) using a quantitative proteomic analysis would strengthen this manuscript and provide some molecular insights about fibroblast-derived factors involved in the reported macrophage activation/polarization.

In summary, the reported findings are very interesting and the studies are well performed. However, in the current format, the molecular mechanisms how FAK activity in fibroblasts impacts tumour cell invasion and/or macrophage polarization remains largely unknown. Thus, an unbiased proteomic based approach to identify some key factors regulating the described changes in tumour cell and/or macrophage functions would clearly strengthen this manuscript.

Referee #2 (Remarks for Author):

Zaghdoudi et al. address the relevance of FAK in PDAC biology and clinical behavior. The topic is relevant since PDAC is associated with a very strong desmoplastic reaction. While the study of the relevance of FAK in cancer, and in PDAC specifically, has been addressed by other investigators, this work focuses mainly on its role in fibroblasts, providing significantly novel information. Overall,

the work is well performed and represents an important addition to the field, although I have a few comments.

Specific comments

1. The specificity of the antibody against p-FAK used is crucial to the interpretation of the findings, therefore the authors should provide convincing evidence using both immunohistochemistry and western blotting. This is particularly important re: the immunostainings in Figure 1 in normal tissue, where it appears that there is staining pretty much all over in normal pancreas.
2. The clinical associations are interesting and are emphasized by the authors. However, they should tone down some of the conclusions since the HR for high pFAK scores are very modest (in the range of 2) and probably of little clinical applicability.
3. Do the authors have any information re: pFAK expression in the stroma of chronic pancreatitis and of PDAC metastatic lesions? this information is not essential for the paper but it would be useful to place the findings in context.
4. The results on the effect of FAK inactivation in fibroblasts on the tumor microenvironment (macrophages) are interesting but some of the changes are modest. In this regard, I find that the data presented in Figure 6 might fit better immediately after Figure 3. Using more than one marker to define M1 and M2 macrophages would be valuable.
5. Figure 5: it is essential that the analysis of matrisome markers presented in panel A is substantiated by western blotting, for two reasons: the differences are small and the data will be more quantitative.
6. The authors should try to put together a model of how FAK inhibitors would impact, in an integrated manner, on the various tumor and microenvironmental components to affect tumor progression/metastasis. The way the data is presented is too fragmentary.

Minor comments

The paper would benefit from some English editing.

Referee #3 (Remarks for Author):

EMM-2020-12010

Fibroblastic FAK Activity is a Prognostic Marker and a Druggable Key Player in Pancreatic Cancer
Sonia Zaghdoudi et al.

This manuscript describes elevated levels of phosphorylated FAK in fibroblasts in pancreatic cancer, which is determined to be an independent marker of poor prognosis. Using an orthotopic, syngeneic model of pancreatic cancer, where tumor cells and fibroblasts are injected orthotopically, FAK activity in fibroblasts is implicated in metastasis of the tumor cells. The authors show that FAK in fibroblasts is required for efficient migration and invasion. Further impairing FAK activity in fibroblasts reduces tumor cell migration. They also demonstrate that pharmacological inhibitions of FAK reduces deposition of extracellular matrix proteins and that FAK activity is required for fibroblast contractility. The authors also suggest that FAK activity in tumor-associated fibroblasts alters the macrophage type present in tumors, and that factors secreted from fibroblasts in the

presence of FAK activity promote the polarization of macrophages into M2 macrophages and promote the migration of macrophages. Overall, the authors conclude that FAK activity in fibroblasts in pancreatic cancer provides a useful prognostic marker, and inhibition of FAK activity might be an effective therapeutic strategy to prevent pancreatic cancer metastasis.

Overall, the study addresses the important problem of prostate cancer and makes the interesting correlation of phosphorylation of FAK in CAFs as a prognostic marker and that FAK activity in fibroblasts is required for metastasis in an orthotopic tumor model. The authors explore mechanism and suggest that very well established functions of FAK are responsible for their observations, i.e. regulation of cell migration and extracellular matrix deposition/organization. These studies are correlative and incremental. An exciting idea is the role of FAK in fibroblasts in promoting polarization of macrophages and recruitment into tumors. This data is a little less convincing, the effect is weak and there is no attempt to define the mechanism. As the authors point out, there are a number of clinical trials utilizing FAK inhibitors in pancreatic cancers. Therefore, this manuscript may have little impact upon treatment of the disease.

Specific critiques include:

- 1) The authors should describe controls to validate the authenticity of their IHC. E.g., describe the controls to validate the pY397 FAK is detecting phosphorylated FAK.
- 2) Some additional controls are required. E.g., in figure 3, do orthotopic tumors (without co-injected fibroblasts) metastasize? In figure 6, do macrophages migrate in serum free medium?
- 3) Some experimental details are missing. E.g., how is the area of metastasis calculated (Fig 3)? How are collagen I and collagen III quantified and what does the scale mean (Fig 3)? How is directionality calculated (Fig 4)? Is distance migrated the displacement from initial point to final point or does it measure the length of the path traveled?
- 4) In Fig 1, quantification of FAK is scored on a scale of 0 to 6, but the Materials and Methods describing scoring on a scale of 1 to 3. Please clarify.
- 5) Standard deviation should be used instead of SEM
- 6) Why is a one-tailed Student's t-test performed on the immunophenotyping data in Fig 3? From Fig 3D, it appears that increases and decreases could be expected and a two-tailed test might be more appropriate.
- 7) In some cases data is normalized and in some cases it is not. How do you account for variability in the control sample when the data is normalized to 100%?
- 8) Additional editing would improve the manuscript.

***** Reviewer's comments *****

Referee #1 (Remarks for Author):

This manuscript shows some exiting data demonstrating a tumour promoting function of high FAK activity in cancer associated fibroblasts (CAFs). The manuscript is structured into two parts. In the first part the authors explore whether the level of FAK activation in cancer associated fibroblasts (CAFs) could be used as a prognostic marker for resected PDAC patients. Therefore, the analysis of two different tissue arrays (TMA) is performed (Figs 1-2). To assess FAK activation, both sets of TMAs were stained using an antibody specific for FAK phosphorylation at tyrosine 397 (pY397), a marker for FAK activation, and fibroblasts were identified based on their morphology using manual scoring or automated image analysis. The authors observe that CAFs show increased FAK activation compared to fibroblasts in tumour free pancreas and that in resected PDAC patients' high levels of FAK activation in CAFs correlates with reduced disease-free survival and reduced overall survival. The analysis is overall well performed and comprehensive.

In the second part, the authors aim to better understand how FAK activation in CAFs promotes PDAC progression by using a variety of in vivo and in vitro experiments. First, the authors use a syngeneic mouse model mouse, whereby KPC derived cancer cells are orthotopically co-injected into the pancreas together with immortalised mouse embryonic fibroblast (MEF) cell lines initially generated from WT and FAK kinase dead (FAK-KD) mutant mice (Fig. 3). This model reveals that tumours containing FAK KD fibroblasts show a reduction in extracellular matrix deposition and a change in tumour associated macrophage polarization towards a pro-inflammatory M1-like phenotype compared to animals co-injected with FAK WT fibroblasts. Most strikingly, co-implantation of FAK KD fibroblasts significantly reduces lung metastasis compared to animals co-implanted with FAK WT fibroblasts. The authors further describe that FAK KD fibroblasts reduce the migration and invasion of tumour cells in vitro (Fig 4) and that pharmacological inhibition of FAK reduces ECM production and deposition of fibroblasts in vitro (Fig 5). In Fig 6, the authors explore the effect of fibroblast conditioned media (CM) collected from fibroblast treated or not with a FAK inhibitor on macrophage activation and migration. They observe that macrophages cultured in CM collected from FAK inhibitor treated fibroblasts show less expression of the M2 marker CD206 and that treatment of fibroblasts with a FAK inhibitor is less potent to attract M2-like macrophages compared ton CM of untreated fibroblasts.

Strength: t

These are all interesting observations describing that high FAK activity in fibroblasts i. correlates with poor prognosis in PDAC patients, ii. promotes ECM production/deposition by fibroblasts, iii. increases cancer cell invasion/metastasis and iv. attracts M2-like macrophages to the site of tumour formation. These findings are novel since the role of FAK in PDAC progression has mainly been studied with a focus on cancer cells and not on the surrounding stroma compartment. The manuscript is well structured and the data is well presented.

Main concerns:

Fig 2B-E: Please explain in more detail how the survival curves displayed in Fig2B-E were generated. The method section describes that patients were scored based on neg/low/med/high pY397 FAK levels as shown in Fig2A. However, the graphs in panel B-E show only two groups (low/high), with n=60 patients each. Were there no patients with medium pY397 FAK levels? Or, do the graphs as displayed compare

the upper half versus the lower half of pY397 FAK levels based on the median? Please explain.

We agree with reviewer 1 and apologize for these missing information. As suggested by the reviewer, we have now included details regarding how the two groups were generated in the Materials and Methods section – Patient Samples: “*Statistical analyses: Percentage of cells expressing low, medium and high pY397 FAK staining were quantified using Definiens Tissue Studio® (Imag’IN core Institut Universitaire du Cancer, Toulouse, France) in stromal and tumour areas. pY397 FAK scoring (h-score) was calculated as followed: 1X%low pY397 FAK cells + 2X%medium pY397 FAK cells +3X%high pY397 FAK cells. Patients were separated in half (60/60) based on their pY397 FAK h-score (either in CAFs or in tumour cells depending on the performed analysis), generating pY397 FAK low group (lower half based on the median) and pY397 FAK high group (upper half) of 60 patients each. The outcome variables were disease free survival (DFS) and overall survival (OS)*”.

Fig 3: Characterization of in vivo model: FAK KD reduces collagen production. Does the altered stroma also affect the architecture of the tumour vascularization which could also impact metastatic spreading? Include stain for vessel density/vessel maturation and hypoxia. Along these lines, since FAK deficient CAFs reduce tumour cell invasion as reported in Fig 4J, please also check for changes in cancer cell invasion at the invasive front of these tumours.

As suggested, we have performed IHC analyses in order to better characterize the impact of fibroblastic FAK inactivation on tumour vessels and hypoxia. Staining and quantification of CD31 IHC are now included in the manuscript (figure 3D). To deeper analyze the vasculature, we planned to quantify vessels coverage with pericytes by performing co-IF of pericyte marker (NG2) and endothelial cell marker (MECA32). Briefly, paraffin-embedded tumour slices were stained with anti-NG2 (neuron-glia antigen 2) and anti-MECA32 (plasmalemma vesicle-associated protein) antibodies and DAPI. Unfortunately, as shown in figure bellow, pericytes and MEFs express NG2 at similar levels, making impossible the pericyte coverage quantification specifically in pericyte/vessels. As other markers such as α SMA and PDGFR for examples are also expressed by MEFs, we were not able to deeper analyze the tumor vasculature maturation, and apologize for that.

In order to analyze hypoxia, we performed IHC staining of hypoxia inducible factor alpha (HIF1a) and its target CAIX (figure 3E). Quantification shows a slight decrease of hypoxia markers in the FAK-KD fib. tumour group (but not significant).

We really want to thank reviewer 1 for his question about changes in cancer cell invasion at invasive front as the results (now included in figure 6A and EV5) have strengthened the manuscript. Indeed, we identify a drastic difference between the two groups: *“Figure 6A shows that, in the FAK-WT fibroblast tumours, CAFs are oriented toward the adjacent pancreatic tissue, thus with a radial arrangement (parallel to the tumour radius but aligned perpendicularly to the tumour edge), but FAK-KD CAFs are organized perpendicular to the tumour radius (parallel to the tumour edge). Interestingly, ECM (Sirius Red) and CAFs have the same orientation (Figures 6A, EV5A).”*

Fig 4: The described reduced directional migration of tumour cells in the presence of FAK KD fibroblasts compared to tumour cells co-culture with FAK WT fibroblasts is very interesting, but the underlying molecular mechanisms remain largely elusive.

As mentioned by the authors, fibroblasts provide invasive tracks made of ECM deposition for tumour cells. Thus, the authors could explore in more detail the molecular composition of these tracks and how FAK activity changes their deposition and ultimately the migration capacity of tumour cells (different integrins involved).

Based on reviewer comment, we first characterized the impact of FAK activity in ECM organization. To do so, we first performed the scratch wound healing assay using activated fibroblasts (FAK-WT or FAK-KD). Upon wound closure, decellularization was performed and ECM stained using NHS-alexa 488 (Figure EV5B). Analysis of deposited ECM during fibroblast migration shows that FAK activity is required for the production of ECM tracks as activated FAK-WT fibroblasts generate dense and abundant ECM fibers, oriented through the wound, whereas FAK-KD fibroblasts do not deposit ECM tracks but ECM heaps (Figure 6B). In order to better characterize the difference in ECM composition, we performed matrisome analyses of deposited matrix from human primary CAFs treated or not by FAK inhibitor (see below, point Fig 5, and figure 6 F to J and EV5 H to J). Finally, we also analyzed the impact of FAK-I induced matrix modifications on tumour cell integrin beta 1 activation. As now shown in Figure 6L and EV5K, $\beta 1$ integrin activation from tumor cells is dramatically decreased upon adhesion to ECM deposited from FAK-I treated hCAFs compared to non-treated hCAFs (Figures 6L, EV5K), while $\beta 1$ integrin expression is not changed.

Fig 5: The analysis of ECM production is largely based on IF staining's. The authors should use an additional method to confirm the observed changes in ECM production. The manuscript would benefit of a more comprehensive analysis beside the selected proteins. For example, the authors could apply a proteomic based secretome or ECM deposition analysis to thoroughly assess the function of FAK in ECM protein secretion and or deposition.

As suggested, we performed an unbiased proteomic based approach named “matrisome” on 3 different human primary CAFs. Data are available via ProteomeXchange with identifier PXD018899 (Username: reviewer60110@ebi.ac.uk Password: Xg5PyR2a).

Figure 6 (F-K) and EV5 (H-J) clearly validate our claim that FAK inactivation in CAFs downregulates expression of core matrisome proteins. Indeed *“This analysis shows that FAK inhibition strongly decreases the expression of proteins belonging to the “core matrisome” (structural and fibrillar extracellular components, glycoproteins and proteoglycans) which drops from 23% to 16% of whole matrisome content. In contrast, “matrisome associated proteins” (ECM regulators, ECM affiliated*

proteins and ECM-secreted factors) are globally not impacted by FAK inhibition (Figure 6F). Figures 6I-L show that, among the 3 hCAFs tested, two present, under FAK inactivation, a strong decrease of collagens (10 different collagens were detected, Figure 6G), proteoglycans (3 detected, Figure 6H) and glycoproteins (40 proteins detected, Figure 6I,J). Importantly, as opposed to CAF1 and 2, basal level of collagens, proteoglycans and glycoproteins produced by CAF3 are very low and are not impacted by FAK inhibition (Figures 6G-I). Such differences in term of response to treatment are in accordance with our IF results (Figure EV5F) and were expected as recent publications have reported CAF heterogeneity in PDAC tumours (36). Regarding “matrisome associated proteins” that are globally not modified by FAK inhibition (Figure 6K), ECM regulators (32 detected) are downregulated in all 3 CAFs whereas ECM affiliated proteins (affiliated either structurally or physically with the core matrisome, 14 detected) and secreted factors (such as growth factors and cytokines, known or suspected to bind to ECM, 17 detected) show no modification (Figures 6K, EV5H-J).”

Fig 6: Changes in macrophage polarization is assessed based on the two markers CD206 and Nos2. The authors should assess additional M1/M2-like markers to better characterise changes in their activation. Also, as written it appears that murine macrophages were used to assess the effect of human CAF conditioned media on macrophage activation. Some of the secreted factors might not conserve their biological activity across the different species. Also, a more detailed analysis of the fibroblast secretome (+/- FAK inhibitor) using a quantitative proteomic analysis would strengthen this manuscript and provide some molecular insights about fibroblast-derived factors involved in the reported macrophage activation/polarization.

We agree on reviewer comment and performed additional M1/M2-like markers to better characterize the impact of CAF-secreted factors on their activation. As recommended, conditioned medium from pre-activated mouse fibroblasts (pre-activation was performed by treating fibroblasts during 48h with mouse tumour cell secreted factors) was used on mouse BMDM. Results are now included in the revised version of the manuscript (Figure 4C-D) and include the analyses of the expression of CD80, CD86, Nos2, CMH2, CD206, dectin and CD163 markers.

In order to identify factors involved in the CAF-induced macrophage polarization/migration, RT-qPCR of CAFs treated or not with FAK inhibitor were performed on specific chemokines produced by stroma and tumor cells and involved in the macrophage recruitment to the tumor. Besides M-CSF, the CC chemokines CCL2, CCL3, CCL4, and CCL5 are well-recognized chemotactic factors for macrophage populations (Lahmar Q et al. *Biochimica et biophysica acta*. 2016;1865(1):23-34; Ruytinx P et al. *Frontiers in immunology*. 2018;9:1930). Results are now included in the revised manuscript: *“mRNA expression levels of M-CSF, CCL2, CCL3, CCL4 and CCL5 were analyzed in four hCAFs treated with FAK-I. CCL3 and 4 mRNA were undetectable. M-CSF and CCL5 mRNA expression is not altered by FAK-I treatment, however CCL2 is drastically reduced both at mRNA (59.4% of decrease +/- 12.5) (Figure 4F) and protein level (84.9% of decrease +/- 3.9) (evaluated by Elisa in hCAF CM) compared to untreated hCAFs (Figure 4G).”*

In summary, the reported findings are very interesting and the studies are well performed. However, in the current format, the molecular mechanisms how FAK activity in fibroblasts impacts tumour cell invasion and/or macrophage polarization remains largely unknown. Thus, an unbiased proteomic based approach to identify some key factors regulating the described changes in tumour cell and/or

macrophage functions would clearly strengthen this manuscript.

Referee #2 (Remarks for Author):

Zaghdoudi et al. address the relevance of FAK in PDAC biology and clinical behavior. The topic is relevant since PDAC is associated with a very strong desmoplastic reaction. While the study of the relevance of FAK in cancer, and in PDAC specifically, has been addressed by other investigators, this work focuses mainly on its role in fibroblasts, providing significantly novel information. Overall, the work is well performed and represents an important addition to the field, although I have a few comments.

Specific comments

1. The specificity of the antibody against p-FAK used is crucial to the interpretation of the findings, therefore the authors should provide convincing evidence using both immunohistochemistry and western blotting. This is particularly important re: the immunostainings in Figure 1 in normal tissue, where it appears that there is staining pretty much all over in normal pancreas.

We agree on reviewer comment and performed experiment to validate the specificity of the antibody against p-Y397 FAK. Data are now included in the manuscript in Appendix figure S1.

We used two approaches: one uses lambda phosphatase to show that the antibody is phosphospecific, and the other uses a FAK phosphopeptide to show that the antibody is specific to the sequence that it's supposed to recognize.

Indeed, as the antibody recognizes the 390 – 401aa of Uniprot# Q05397, we used a blocking phosphospecific peptide from abcam (ab40145).

Both the blocking phosphospecific peptide and the lambda phosphatase totally abrogate the IHC staining as well as induce the disappearance of the 125 kDa band in western blot, altogether validating the specificity of the antibody used. Data are now included in the revised version in Appendix figure 1

2. The clinical associations are interesting and are emphasized by the authors. However, they should tone down some of the conclusions since the HR for high pFAK scores are very modest (in the range of 2) and probably of little clinical applicability.

We respectfully disagree with reviewer, as for example CA19-9 serum levels and lymph node ratio (LNR), classically used in clinic and shown to be correlated with PDAC patient prognosis, have HR in a range of 2. Indeed, regarding CA19-9, Hammad N et al. reported, in a clinical study analyzing 111 patients with pancreas cancer, that “lower baseline CA19-9 levels were positively associated with OS (median 9.1 vs 6.1 months, $P = 0.0057$) and TTP (Time to progression) (median 6.4 vs 4.2 months, $P = 0.0044$). The covariate adjusted hazard ratio (HR) for progression among patients with baseline CA19-9 ≥ 1000 ng/mL was **HR = 1.94** (95% CI 1.24-3.02), with $P = 0.0035$. The covariate adjusted risk of death among patients with baseline CA19-9 ≥ 1000 ng/ml was similarly elevated: HR = 1.90 (95% CI 1.23-2.94), with $P = 0.0039$ ” (Hammad N et al Asia Pac J Clin Oncol. 2010 Jun;6(2):98-105. doi: 10.1111/j.1743-7563.2010.01290.x.). Moreover, Boeck et al. demonstrated that CA 19-9 kinetics was a significant predictor of both time to tumour progression (Hazard Ratio, **HR 1.48**, $p < 0.001$) and overall survival (HR 1.34, $p < 0.001$) in a cohort of 115 patients (first line pancreatic cancer chemotherapy). Authors conclude

that “Pretreatment CA 19-9 and CA 19-9 kinetics may serve as a useful serum biomarker in advanced pancreatic cancer”. (Boeck S et al. Clin Cancer Res. 2010;16(3):986–994. doi: 10.1158/1078-0432.CCR-09-2205). CA19-9 serum levels can provide important information with regards to prognosis, overall survival, and response to chemotherapy as well as predict post-operative recurrence. According to Bartel et al., numerous other potential markers have been tested and compared against CA19-9, but none has proven more accurate (M.J. Bartel, ... M. Raimondo, in Diagnostic Molecular Pathology, 2017). Lymph Node Ratio (LNR) is, according to Riediger et al., the strongest prognostic factor after resection of PDAC with **HR=1.6 or 2.2** depending on cutoff (J Gastrointest Surg. 2009). The relationship between LNR and PDAC patient overall survival was evaluated by Elshaer et al. through a systematic review of 19 studies including 4,883 patients, and 17 studies showed that a high LNR was associated with decreased overall survival with HR comprised between **1.5 and 2.8** (Elshaer et al., Ann R Coll Surg Engl. 2017 Feb; 99(2): 101–106). While we agree that analyzing FAK activity specifically in fibroblasts could be challenging, we want to point out that there is now software available allowing a reproducible and sensitive quantification of IHC staining (free software : QuPath; commercial software: Definiens).

Altogether, we believe that an HR around 2 is not modest in PDAC and highlights the potential benefit of evaluating FAK activity within CAFs to identify patients with high risk of early relapse and patients that will benefit from personalized protocol using FAK inhibitor treatment.

3. Do the authors have any information re: pFAK expression in the stroma of chronic pancreatitis and of PDAC metastatic lesions? this information is not essential for the paper but it would be useful to place the findings in context.

We thank reviewer for such question but apologize as we have no answer so far. Access to patient metastatic tissue is difficult since metastatic patients are not operated. Cytopunction material is not sufficiently informative to perform reliable IHC. Patient pancreatitis tissue samples could inform on whether FAK is activated in non-cancerous inflammatory lesions where fibroblasts are already activated into myofibroblasts, and this may be tested in a future study. In mice, However, Hong Jiang et al. publication (Nat Med 2016) Figure 1F shows IHC staining of p-FAK1 in pancreatic tissue from the p48-Cre/LSL-KrasG12D/p53Flox/+ (KPC) mouse model, where FAK activity seems not induced in fibroblasts around late Panin lesions.

4. The results on the effect of FAK inactivation in fibroblasts on the tumor microenvironment (macrophages) are interesting but some of the changes are modest. In this regard, I find that the data presented in Figure 6 might fit better immediately after Figure 3. Using more than one marker to define M1 and M2 macrophages would be valuable.

We thank reviewer for this comment. All data regarding immune cells are now included in Figure 4 and EV3. We also repeated the CM-induced macrophage polarization experiment using mouse BMDM and CM from mouse fibroblasts. We analyzed for a total of 8 different macrophage markers (CD163, CD206, F4/80, CD11b, CD80, CD86, dectin, MHCII), and show that CM from CAFs favors M2 macrophage polarization dependently on FAK activity. Figure 4D shows that Dectin and CD206 markers are decreased when macrophages are stimulated with CM from FAK-KD activated fibroblasts. This two markers are mainly expressed by the M2a macrophage sub-group, which is of interest as M2a macrophages are known to be profibrotic.

5. Figure 5: it is essential that the analysis of matrisome markers presented in panel A is substantiated by western blotting, for two reasons: the differences are small and the data will be more quantitative.

In order to answer reviewer comment, western blots on primary human CAFs and on tumour lysates were carried out. We confirm coll I, III, and periostin decreases upon FAK inactivation in both in vitro CAFs and in tumors, whereas coll IV and OPN decreases are more evident in tumors (representative WB are shown in figure EV5G).

In addition, we performed an unbiased proteomic based approach named “matrisome” on 3 different human primary CAFs allowing a quantitative analysis of the matrix. Data are available via ProteomeXchange with identifier PXD018899 (Username: reviewer60110@ebi.ac.uk Password: Xg5PyR2a).

Figure 6 (F-K) and EV5 (H-J) clearly validate our claim that FAK inactivation in CAFs downregulates expression of core matrisome proteins. Indeed “This analysis shows that FAK inhibition strongly decreases the expression of proteins belonging to the “core matrisome” (structural and fibrillar extracellular components, glycoproteins and proteoglycans) which drops from 23% to 16% of whole matrisome content. In contrast, “matrisome associated proteins” (ECM regulators, ECM affiliated proteins and ECM-secreted factors) are globally not impacted by FAK inhibition (Figure 6F). Figures 6I-L show that, among the 3 hCAF tested, two present, under FAK inactivation, a strong decrease of collagens (10 different collagens were detected, Figure 6G), proteoglycans (3 detected, Figure 6H) and glycoproteins (40 proteins detected, Figure 6I,J). Importantly, as opposed to CAF1 and 2, basal level of collagens, proteoglycans and glycoproteins produced by CAF3 are very low and are not impacted by FAK inhibition (Figures 6G-I). Such differences in term of response to treatment are in accordance with our IF results (Figure EV5F) and were expected as recent publications have reported CAF heterogeneity in PDAC tumours (Neuzillet et al. The Journal of pathology. 2019;248(1):51-65) which is also observed for fibroblastic pY397 FAK in different patients. Regarding “matrisome associated proteins” that are globally not modified by FAK inhibition (Figure 6K), ECM regulators (32 detected) are downregulated in all 3 CAFs whereas ECM affiliated proteins (affiliated either structurally or physically with the core matrisome, 14 detected) and secreted factors (such as growth factors and cytokines, known or suspected to bind to ECM, 17 detected) show no modification (Figures 6K, EV5H-J).”

6. The authors should try to put together a model of how FAK inhibitors would impact, in an integrated manner, on the various tumor and microenvironmental components to affect tumor progression/metastasis. The way the data is presented is too fragmentary.

As asked by reviewer, we have added a model of how fibroblastic FAK activity impacts tumour microenvironment (ECM, secreted factors and immune cells) leading

to tumour progression. It is now included in the “the paper explained” section.

Minor comments

The paper would benefit from some English editing.

English editing has been performed by English-native researcher.

Referee #3 (Remarks for Author):

EMM-2020-12010

Fibroblastic FAK Activity is a Prognostic Marker and a Druggable Key Player in Pancreatic Cancer

Sonia Zaghdoudi et al.

This manuscript describes elevated levels of phosphorylated FAK in fibroblasts in pancreatic cancer, which is determined to be an independent marker of poor prognosis. Using an orthotopic, syngeneic model of pancreatic cancer, where tumor cells and fibroblasts are injected orthotopically, FAK activity in fibroblasts is implicated in metastasis of the tumor cells. The authors show that FAK in fibroblasts is required for efficient migration and invasion. Further impairing FAK activity in fibroblasts reduces tumor cell migration. They also demonstrate that pharmacological inhibitions of FAK reduces deposition of extracellular matrix proteins and that FAK activity is required for fibroblast contractility. The authors also suggest that FAK activity in tumor-associated fibroblasts alters the macrophage type present in tumors, and that factors secreted from fibroblasts in the presence of FAK activity promote the polarization of macrophages into M2 macrophages and promote the migration of macrophages. Overall, the authors conclude that FAK activity in fibroblasts in pancreatic cancer provides a useful prognostic marker, and inhibition of FAK activity might be an effective therapeutic strategy to prevent pancreatic cancer metastasis.

Overall, the study addresses the important problem of prostate cancer and makes the interesting correlation of phosphorylation of FAK in CAFs as a prognostic marker and that FAK activity in fibroblasts is required for metastasis in an orthotopic tumor model. The authors explore mechanism and suggest that very well established functions of FAK are responsible for their observations, i.e. regulation of cell migration and extracellular matrix deposition/organization. These studies are correlative and incremental. An exciting idea is the role of FAK in fibroblasts in promoting polarization of macrophages and recruitment into tumors. This data is a little less convincing, the effect is weak and there is no attempt to define the mechanism. As the authors point out, there are a number of clinical trials utilizing FAK inhibitors in pancreatic cancers. Therefore, this manuscript may have little impact upon treatment of the disease.

Specific critiques include:

1) The authors should describe controls to validate the authenticity of their IHC. E.g., describe the controls to validate the pY397 FAK is detecting phosphorylated FAK.

We agree on reviewer comment and performed experiment to validate the specificity of the antibody against p-Y397 FAK. Data are now included in the manuscript in Appendix figure S1.

We used two approaches: one uses lambda phosphatase to show that the antibody is phosphospecific, and the other uses a FAK phosphopeptide to show that the antibody is specific to the sequence that it's supposed to recognize.

Indeed, as the antibody recognizes the 390 – 401aa of Uniprot# Q05397, we used a blocking phosphospecific peptide from abcam (ab40145).

Both the blocking phosphospecific peptide and the lambda phosphatase totally abrogate the IHC staining as well as induce the disappearance of the 125 kDa band in western blot, altogether validating the specificity of the antibody used. Data are now included in the revised version in Appendix figure 1

2) Some additional controls are required. E.g., in figure 3, do orthotopic tumors (without co-injected fibroblasts) metastasize?

According to reviewer comment, Figure 3G now encompasses metastasis quantification in mouse grafted with tumour cells alone (as well as, in Figure 3A, tumor mass), which show that co-grafting tumor cells with FAK-WT fibroblasts, but not FAK-KD significantly increases metastasis number and area. Regarding tumor mass, we show that, compared to tumor cells alone, the co-grafting of tumor cells and fibroblasts (either FAK-WT or FAK-KD) increase tumor mass (but not significantly).

In figure 6, do macrophages migrate in serum free medium?

Figure EV3G shows quantification of M1 and M2 macrophages in DMEM/F12 + 0.5% FBS (identified "M" for medium): 2% of M1 macrophages and 7% of M2 are able to migrate through the transwell membrane, this migration being significantly increased for both M1 and M2 macrophages by MC from CAFs, but not from CAFs pre-treated with FAK-I.

3) Some experimental details are missing. E.g., how is the area of metastasis calculated (Fig 3)? How are collagen I and collagen III quantified and what does the scale mean (Fig 3)? How is directionality calculated (Fig 4)? Is distance migrated the displacement from initial point to final point or does it measure the length of the path traveled?

We apologize for such omission. Details are now included in the material and method section or in figure legend:

Material and method:

- *"Areas of lung metastasis were quantified based on CK19 staining allowing tumour cell identification. Percentage of metastasis area is the ratio of CK19 positive area divided by total lung area * 100."*
- *"According to manufacturer's instructions (Abcam, ab150681), areas of yellow-orange birefringent fibers corresponding to type I (thick fibers) collagen, and green birefringent fibers corresponding to type III (Thin fibers) collagen, were quantified using ImageJ."*
- *"Distance of migration (length of the path traveled), directionality (calculated by comparing the Euclidian distance to the accumulated distance, represents a measurement of the directness of cell trajectories) and migration velocity (rapidity of cell motion, calculated on moving cells) were analyzed using Chemotaxis and Migration Tool software (free software tool for data analysis from time stack chemotaxis experiments, based on the National Institute of Health's (NIH) ImageJ image processing system)."*

Scale bars are now included in figure legends

4) In Fig 1, quantification of FAK is scored on a scale of 0 to 6, but the Materials and Methods describing scoring on a scale of 1 to 3. Please clarify.

Clarification has been added in the materials and Methods:

"Two spots per patient were double blinded quantified based on pY397 FAK staining intensity and positive

cell number (a total of 4 quantifications were performed per patient, and mean calculated). pY397 FAK cytoplasmic and nuclear staining were both scored on a scale of 6 : 0 for no staining and 6 for high positive staining, taking into account staining intensity and positive cell quantity. pY397 FAK CAF staining is the mean of cytoplasmic and nuclear score."

5) Standard deviation should be used instead of SEM

According to Embo Molecular Medicine guidelines regarding statistical analysis: "Descriptive statistics should include a clearly labelled measure of centre (such as the mean or the median), and a clearly labelled measure of variability (such as standard deviation or range). [...] Authors must state whether a number that follows the \pm sign is a standard error (s.e.m.) or a standard deviation (s.d.) Authors must justify the use of a particular test and explain whether their data conform to the assumptions of the tests.", we have thus decided to keep the SEM. However, if there is a scientific reason to ask for SD instead of SEM that we don't know, please let us know as we will reconsider our choice.

6) Why is a one-tailed Student's t-test performed on the immunophenotyping data in Fig 3? From Fig 3D, it appears that increases and decreases could be expected and a two-tailed test might be more appropriate.

We agree on reviewer comment. However, we can explain our choice of using one-tailed test (that we changed for a two-tailed test). Indeed, at the time when we performed the immunophenotyping experiment, we already knew that high fibroblastic pY397 score in PDAC patient samples correlated with CD206+ tumour-associated macrophage abundance (Figure 4H), and thus, we expected a decrease of the M2 macrophage number in tumours with FAK-KD fibroblasts.

As immunophenotyping comes first in the manuscript (Figure 4A), we have repeated the experiment to increase mice number, in order to verify the significance of our results using two-tailed test. As shown in Figure 4A, M2 macrophage number within FAK-KD fib. tumours is significantly decreased at early and late time points.

7) In some cases data is normalized and in some cases it is not. How do you account for variability in the control sample when the data is normalized to 100%?

We understand reviewer comment, and performed modification in figure 4D allowing to see the impact of CM from control sample on macrophage polarization. However, regarding figure 6D and EV5F, that encompass up to 11 different hCAFs, we choose to keep results normalized on control samples due to important CAF heterogeneity that complexifies the figure without adding information. We believe that normalizing on control samples facilitates figure reading and understanding, and, importantly, as all raw data (such as MFI of IF staining) will be included, readers would have access to information such as variability in control samples.

8) Additional editing would improve the manuscript.

Additional editing has been done.

14th Aug 2020

Dear Dr. Jean,

Thank you for the submission of your revised manuscript to EMBO Molecular Medicine, and please accept my apologies for the delay in getting back to you. We have now received the enclosed reports from the two referees who reviewed the new version of your manuscript. As you will see, they are now overall supportive of publication. Still, referee #2 raises a few minor concerns that should be addressed experimentally and in writing. Furthermore, before acceptance, please address the following editorial amendments:

1) Main manuscript text:

- Please answer/correct the changes suggested by our data editors in the main manuscript file (in track changes mode). This file will be sent to you in the next couple of days. Please use this file for any further modification.
- Please remove the paragraph "Significance".
- Author contributions: the contribution of every author must be detailed in a separate section (before the acknowledgments). Please also add Manon Strehaiano in the submission system.
- Please note that all corresponding authors are required to supply an ORCID ID. An ORCID identifier is missing for Corinne Bousquet.
- Thank you for providing "The Paper Explained". Please move this section before the references.
- References should be listed in alphabetical order and 10 authors should be listed before "et al".
- Funding: LNCC fellowship (IP/SC16060), (INCa, PLBIO2015-115, FRM (#40493) and Ligue Nationale Contre le Cancer (LNCC) are in the manuscript but not in the submission system. Please merge with the Acknowledgements.
- Thank you for including a Data Availability section. Please note that the data is not yet accessible and should be made public before acceptance of the manuscript.
- The main text mentions Fig. 4I, but there is no such panel. References for all panels of Fig. EV1 are missing in the manuscript text.
- In the material and methods section: Please include a statement confirming that informed consent was obtained from all subjects (missing for 120 patient cohort) and that the experiments conformed to the principles set out in the WMA Declaration of Helsinki and the Department of Health and Human Services Belmont Report.
- Please indicate in the legends exact $n=$ and exact $p=$ values, not a range, along with the statistical test used. Some people found that to keep the figures clear, providing a supplemental table with all exact p -values was preferable. You are welcome to do this if you want to.

2) Figures:

- Please provide a scale bar for the magnification panels in Fig. 1B.
- Dataset EV legends: the movie files need to be renamed "Movie EV1" etc. and need their legends removed from the manuscript and zipped with the respective movie files. The legends for EV Tables should be removed from the main manuscript.
- Table 1 is uploaded as a separate file, please add it to the main manuscript text.

3) Appendix: please merge the 3 appendix figures into 1 file (PDF format), and add a table of content, as well as the figure legends that should be removed from the main manuscript.

4) Source Data: Thank you for providing source data files. Please upload them so as to have one file per figure.

5) Every published paper now includes a 'Synopsis' to further enhance discoverability. Synopses are displayed on the journal webpage and are freely accessible to all readers. They include a short stand first (maximum of 300 characters, including space) as well as 2-5 one-sentences bullet points that summarizes the paper. Please write the bullet points to summarize the key NEW findings. They should be designed to be complementary to the abstract - i.e. not repeat the same text. We encourage inclusion of key acronyms and quantitative information (maximum of 30 words / bullet point). Please use the passive voice. Please attach these in a separate file or send them by email, we will incorporate them accordingly.

Please also suggest a striking image or visual abstract to illustrate your article as a png or jpeg file 550 px-wide x 400-px high. You are welcome to use the nice figure provided in "The Paper Explained" section, as this section does not contain figures.

6) As part of the EMBO Publications transparent editorial process initiative (see our Editorial at <http://embomolmed.embopress.org/content/2/9/329>), EMBO Molecular Medicine will publish online a Review Process File (RPF) to accompany accepted manuscripts.

In the event of acceptance, this file will be published in conjunction with your paper and will include the anonymous referee reports, your point-by-point response and all pertinent correspondence relating to the manuscript. Let us know whether you agree with the publication of the RPF and as here, if you want to remove or not any figures from it prior to publication.

I look forward to receiving your revised manuscript.

Yours sincerely,

Lise Roth

Lise Roth, PhD
Editor
EMBO Molecular Medicine

To submit your manuscript, please follow this link:

Link Not Available

Graphs 800-1,200 DPI
Photos 400-800 DPI
Colour (only CMYK) 300-400 DPI"

*Additional important information regarding figures and illustrations can be found at <http://bit.ly/EMBOPressFigurePreparationGuideline>

The system will prompt you to fill in your funding and payment information. This will allow Wiley to send you a quote for the article processing charge (APC) in case of acceptance. This quote takes into account any reduction or fee waivers that you may be eligible for. Authors do not need to pay any fees before their manuscript is accepted and transferred to our publisher.

***** Reviewer's comments *****

Referee #1 (Comments on Novelty/Model System for Author):

This is a revised version of a previous submitted manuscript. The authors added more mechanistical insights on how FAK expression in cancer associated fibroblasts promotes PDAC proregrression. The findings are novel and timely.

Referee #1 (Remarks for Author):

The authors addressed well each of my previous comments. The quality of the manuscript markedly improved and the findings are novel, interesting and important. Congratulation to the authors for this work.

Referee #2 (Comments on Novelty/Model System for Author):

This is a high quality paper that reports on the effects of fibroblastic FAK on tumor progression; interesting topic given that most people focus on FAK in tumor cells and that there are drugs targeting FAK.

Referee #2 (Remarks for Author):

The authors have responded to most of my comments and the paper is now improved. However, I still have some concerns on the response to my previous points 1 and 2. It would be nice if the authors could perform one additional experiment (point 1) and make some minor changes in the text (point 2), following the arguments I provide below.

Specific comments

1. The results shown on the analysis of antibody specificity are convincing in themselves. However, using a phospho-peptide for inhibition is not adequate: if there is a cross-reactive epitope, the cross-reactivity will be inhibited by the peptide. The way to show that the antibody is specific is a knockdown or CRISPR-knockout.

2. I also respectfully disagree with the authors. CA19.9 levels, as a marker of outcome, have a HR of 2... but this conclusion is made after years of use in clinical practice. In fact, CA19.9 is used because there is nothing better so far - but it is not very useful. By contrast, the value of new markers as indicators of outcome systematically decreases when many studies are performed as follow-up. In biomarker research, the usefulness of a marker essentially never improves as more studies are performed. Therefore, the comparison is unfair and I think that the authors should tone-down their conclusions. I think that this paper merits publication but there is no need to "oversell" anything, for the sake of rigorous science. A HR of 2 is modest and generally of only limited clinical use in decision-making, especially if it refers to selecting a drug for therapy.

***** Reviewer's comments *****

Referee #2 (Remarks for Author):

The authors have responded to most of my comments and the paper is now improved. However, I still have some concerns on the response to my previous points 1 and 2. It would be nice if the authors could perform one additional experiment (point 1) and make some minor changes in the text (point 2), following the arguments I provide below.

Specific comments

1. The results shown on the analysis of antibody specificity are convincing in themselves. However, using a phospho-peptide for inhibition is not adequate: if there is a cross-reactive epitope, the cross-reactivity will be inhibited by the peptide. The way to show that the antibody is specific is a knockdown or CRISPR-knockout.

In order to complete our antibody specificity validation shown in Appendix Figure 1, we performed western blot experiment on FAK deleted cells (MEFs isolated from murine homozygous *Fak* floxed embryos and treated with an adenovirus expressing the cre-recombinase (adeno-cre)). As now shown in the Appendix Figure 1C, FAK deletion induces the disappearance of the 125 kDa band in western blot, altogether validating the specificity of the antibody used.

2. I also respectfully disagree with the authors. CA19.9 levels, as a marker of outcome, have a HR of 2... but this conclusion is made after years of use in clinical practice. In fact, CA19.9 is used because there is nothing better so far - but it is not very useful. By contrast, the value of new markers as indicators of outcome systematically decreases when many studies are performed as follow-up. In biomarker research, the usefulness of a marker essentially never improves as more studies are performed. Therefore, the comparison is unfair and I think that the authors should tone-down their conclusions. I think that this paper merits publication but there is no need to "oversell" anything, for the sake of rigorous science. A HR of 2 is modest and generally of only limited clinical use in decision-making, especially if it refers to selecting a drug for therapy.

We would like to thank the Reviewer for this very interesting exchange. As we agree on the fact that taking into account the "systematic decrease" of the value of new prognostic markers, we carefully edited our manuscript to avoid to oversell our results.

8th Sep 2020

Dear Dr. Jean,

Thank you for submitting your revised version of the manuscript. I have now looked at everything and all is fine. I am therefore very pleased to accept your manuscript for publication in EMBO Molecular Medicine!

It will now be sent to our publisher to be included in the next available issue of EMBO Molecular Medicine.

Please read below for additional important information regarding your article, its publication and the production process.

Congratulations on a nice study!

Sincerely,

Lise Roth

Lise Roth, Ph.D
Scientific Editor
EMBO Molecular Medicine

Follow us on Twitter @EmboMolMed
Sign up for eTOCs at embopress.org/alertsfeeds

*** ** IMPORTANT INFORMATION ** **

SPEED OF PUBLICATION

The journal aims for rapid publication of papers, using the advance online publication "Early View" to expedite the process: A properly copy-edited and formatted version will be published as "Early View" after the proofs have been corrected. Please help the Editors and publisher avoid delays by providing e-mail address(es), telephone and fax numbers at which author(s) can be contacted.

Should you be planning a Press Release on your article, please get in contact with embomolmed@wiley.com as early as possible, in order to coordinate publication and release dates.

LICENSE AND PAYMENT:

All articles published in EMBO Molecular Medicine are fully open access: immediately and freely

available to read, download and share.

EMBO Molecular Medicine charges an article processing charge (APC) to cover the publication costs. You, as the corresponding author for this manuscript, should have already received a quote with the article processing fee separately. Please let us know in case this quote has not been received.

Once your article is at Wiley for editorial production you will receive an email from Wiley's Author Services system, which will ask you to log in and will present you with the publication license form for completion. Within the same system the publication fee can be paid by credit card, an invoice, pro forma invoice or purchase order can be requested.

Payment of the publication charge and the signed Open Access Agreement form must be received before the article can be published online.

PROOFS

You will receive the proofs by e-mail approximately 2 weeks after all relevant files have been sent to our Production Office. Please return them within 48 hours and if there should be any problems, please contact the production office at embopressproduction@wiley.com.

Please inform us if there is likely to be any difficulty in reaching you at the above address at that time. Failure to meet our deadlines may result in a delay of publication.

All further communications concerning your paper proofs should quote reference number EMM-2020-12010-V3 and be directed to the production office at embopressproduction@wiley.com.

Thank you,

Lise Roth, Ph.D
Scientific Editor
EMBO Molecular Medicine

YOU MUST COMPLETE ALL CELLS WITH A PINK BACKGROUND ↓
PLEASE NOTE THAT THIS CHECKLIST WILL BE PUBLISHED ALONGSIDE YOUR PAPER

Corresponding Author Name: JEAN
Journal Submitted to: EMBO Molecular Medicine
Manuscript Number: EMM-2020-12010V2